# SIMUHOME: A TEMPORAL- AND ENVIRONMENT-AWARE BENCHMARK FOR SMART HOME LLM AGENTS

**Gyuhyeon Seo, Jungwoo Yang, Junseong Pyo,**[*] **Nalim Kim, Jonggeun Lee, Yohan Jo**[†]
Graduate School of Data Science, Seoul National University
{seokh97,jwyang0213,yohan.jo}@snu.ac.kr

## ABSTRACT

We introduce SimuHome, a high-fidelity smart home simulator and a benchmark of 600 episodes for LLM-based smart home agents. Existing smart home benchmarks treat the home as a static system, neither simulating how device operations affect environmental variables over time nor supporting workflow scheduling of device commands. SimuHome is grounded in the Matter protocol[1], the industry standard that defines how real smart home devices communicate and operate. Agents interact with devices through SimuHome's APIs and observe how their actions continuously affect environmental variables such as temperature and humidity. Our benchmark covers state inquiry, implicit user intent inference, explicit device control, and workflow scheduling, each with both feasible and infeasible requests. For workflow scheduling, the simulator accelerates time so that scheduled workflows can be evaluated immediately. An evaluation of 18 agents reveals that workflow scheduling is the hardest category, with failures persisting across alternative agent frameworks and fine-tuning. These findings suggest that SimuHome's time-accelerated simulation could serve as an environment for agents to pre-validate their actions before committing them to the real world. Code and data are available at `https://github.com/holi-lab/SimuHome/`.

## 1 INTRODUCTION

Smart home agents have long been a central research topic for tool agents. Systems such as Amazon Alexa and Google Home are among the earliest tool agents commercialized at scale, yet many everyday household requests remain beyond their reach. Building more capable agents with large language models (LLMs) requires handling several challenges that go beyond simple command execution. The simplest case is an explicit command such as *"Turn on the light"*, but not all user requests are this direct. A user who says *"It feels sticky"* expects the agent to recognize this as a request to reduce humidity and activate a dehumidifier. Even when commands are explicit, agents must respect operational dependencies between device commands. For instance, a robot vacuum cleaner must be powered on before it can be switched to mopping mode. Some requests go further and require coordinating device actions across time. A request such as *"Turn on the kitchen light when the dishwasher finishes"* requires the agent to check the dishwasher's remaining cycle time, calculate the expected completion time, and schedule the light to turn on at that time. In addition to these challenges, agents must track how device actions continuously affect the surrounding environment. Setting an air conditioner to 25°C does not change the room temperature instantly. The temperature drops gradually over several minutes, and the agent must observe these ongoing changes to determine whether the target condition has been reached.

Training and evaluating these capabilities require an interactive environment where agents can call APIs to operate devices and observe the resulting changes to environmental variables over time. Existing smart home benchmarks do not model how device actions continuously affect environmental variables such as temperature and humidity. They also do not enforce operational dependencies or support time-accelerated evaluation of scheduling tasks. Beyond these limitations, imitation learning

---

[*]Work done while the author was an intern from the Department of Information Systems, Hanyang University.
[†]Corresponding author.
[1]https://csa-iot.org/all-solutions/matter/

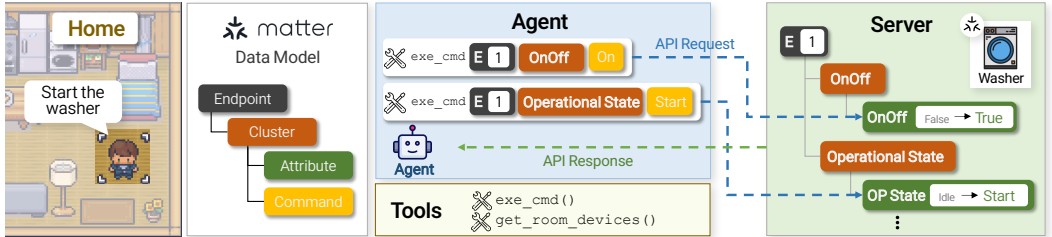

Figure 1: The SimuHome home environment. Agents receive user commands, issue API calls built on the Matter protocol to operate devices, and observe the resulting state changes.

on static datasets is also insufficient because a single user request can often be fulfilled through multiple valid action sequences that fixed annotations cannot cover. Instead, agents must interact with a live environment where their actions produce observable state changes that can be verified against the intended goal. We aim to address this challenge by developing a high-fidelity smart home simulator in which agents can interact with devices through APIs and observe the results reflected in the environment, along with an extensive benchmark containing a variety of complex user requests.

Our first contribution is **SimuHome**, a time-accelerated smart home simulator (Figure 1). Agents interact with the simulator through APIs built on the Matter protocol, a global industry standard that defines how smart home devices communicate and operate. When agents operate devices, the simulator computes continuous changes in environmental variables, allowing agents to observe evolving conditions and decide their next actions. For workflow scheduling, agents can register workflows for future execution, and the simulator accelerates time so that outcomes are verifiable immediately. The simulation is fully reproducible, ensuring fair comparisons across models.

Our second contribution is a manually validated benchmark of 600 episodes built on SimuHome. The benchmark spans six query types, namely state inquiry, implicit user intent inference, explicit device control, and three forms of workflow scheduling. Each query type includes both feasible episodes, where the agent must fulfill the request, and infeasible episodes, where the agent must recognize and explain why the request cannot be fulfilled. In each episode, an agent receives a user query in a configured home environment and is evaluated against a verifiable goal.

Using this benchmark, we evaluate 18 LLM agents and find that while current models handle explicit device control reliably, workflow scheduling remains the most persistent challenge. GPT-5.1, which uses extended chain-of-thought reasoning, improves substantially on workflow scheduling. However, it requires three to five times the inference time, making it impractical for real-time smart home use. We further test alternative agent frameworks and fine-tuning on successful trajectories, but workflow scheduling remains unresolved, suggesting that the bottleneck lies in the models' reasoning capabilities. These findings call for research on agents that combine strong reasoning with the low latency that real-time deployment demands. On the practical side, SimuHome's time-accelerated simulation provides a world model environment where we can extensively train and test deploy-ready smart home agents, which is challenging to conduct in physical homes.

## 2 RELATED WORK

**Embodied Agent Benchmarks in Household Environments.** LLM agents have demonstrated strong tool-use capabilities across diverse domains (Schick et al., 2023; Qin et al., 2024; Patil et al., 2025; Zhou et al., 2024; Xie et al., 2024; Trivedi et al., 2024). In household settings, simulated benchmarks such as AI2-THOR (Kolve et al., 2022), ALFRED (Shridhar et al., 2020), and Virtual-Home (Puig et al., 2018) have advanced instruction following by placing agents in 3D environments where they navigate rooms and manipulate objects through predefined action commands. These benchmarks focus on physical navigation and object manipulation, which are fundamentally different problems from smart home device control, where agents issue API calls to operate networked devices and must reason about their effects on the surrounding environment.

**LLM Agent Benchmarks for Smart Homes.**    To more directly address smart home device control, recent benchmarks evaluate LLM agents on tasks closer to real-world usage. HomeBench (Li et al., 2025) tests instruction following at scale, evaluating agents by comparing their generated API calls against gold-standard sequences across both valid and invalid requests. Sasha (King et al., 2024) focuses on creative goal interpretation, mapping underspecified user intentions such as *"make it cozy"* to device-level action plans, with plan quality assessed through human surveys and automated relevance metrics. SAGE (Rivkin et al., 2024) frames smart home control as sequential tool use, where agents iteratively call APIs, observe outputs, and select the next action. Unlike HomeBench and Sasha, SAGE allows device settings to change dynamically during an agent's execution through the SmartThings API. Despite these advances, none of these benchmarks simulates how device actions affect environmental variables over time, enforces operational dependencies between device commands, or evaluates time-based scheduling. SimuHome addresses these gaps with an interactive simulator grounded in the Matter protocol, with support for time-accelerated scheduling.

## 3    SIMUHOME: A SMART HOME SIMULATOR

### 3.1    MOTIVATION

Building a physical testbed is costly, repeating experiments across diverse configurations is impractical, and waiting in real time for scheduled operations makes large-scale evaluation infeasible. A simulator addresses these challenges, but must closely reflect real device behavior so that agents developed in simulation can transfer to physical smart homes. We design SimuHome around three core requirements.

**Dependency Modeling Based on an Industry Standard.** The simulator must model the operational rules of smart devices according to the Matter protocol, so that device behavior in simulation follows the same constraints as physical devices. For example, operating an air conditioner may require multiple steps in a specific order, such as powering on the device before adjusting its temperature.

**Real-Time Environmental Feedback.** The simulator must model the continuous effects of device operations on environmental variables (e.g., temperature, illuminance, humidity, and air quality). This enables evaluation of whether agents can monitor ongoing changes and react appropriately. For example, as an air conditioner runs, the room temperature gradually decreases toward the target temperature, and the simulator must reflect these changes continuously.

**Workflow Scheduling and Time Acceleration.** The simulator must support workflow scheduling, allowing agents to register a sequence of commands for execution at a specified future time. It must also accelerate simulated time so that the outcomes of scheduled workflows can be observed immediately. This enables evaluation of temporal coordination tasks, such as scheduling a kitchen light to turn on when the dishwasher finishes.

### 3.2    SIMUHOME ARCHITECTURE AND OPERATION

SimuHome divides time into discrete steps called ticks, where each tick represents 0.1 seconds. This tick-based design guarantees fully deterministic state transitions, so that given the same initial conditions and the same sequence of actions, the environment always produces identical outcomes. SimuHome runs in sync with real time by default, but supports time acceleration to quickly reach any future point in the simulation. It comprises three components, each addressing one of the requirements above: the Smart Home Environment defines device configurations and environmental variables, the Real-Time State Update Mechanism computes continuous environmental changes at every tick, and the Agent-Simulator Interface provides the tools through which agents observe and act.

**Smart Home Environment.** A home is a configurable environment composed of one or more rooms, each containing a set of devices and four environmental variables: temperature, illuminance, humidity, and air quality. The environment includes both devices that directly influence environmental variables (e.g., an air conditioner affecting temperature) and those with multi-stage operational cycles (e.g., a washing machine). In total, we model 17 distinct device types. A full list is provided in Appendix B, along with their supported Matter clusters (groups of related device capabilities) in Appendix C. Possible extensions to more complex interactions, such as cross-environment and cross-device effects, are discussed in Appendix D.

**Real-Time State Update Mechanism.** At each tick, the simulator computes the combined influence of all active devices on environmental variables. This influence is additive, scaling with the number of active devices and their settings. For example, running two air conditioners at high fan speed lowers the temperature faster than running one. Sensor attributes on devices, such as a temperature sensor on an air conditioner, are also updated to reflect current environmental variables at each tick. The detailed update equations are provided in Appendix E.

**Agent-Simulator Interface.** Agents do not have direct access to the full environment state. Instead, they interact with the simulator through tools for querying device states and environmental variables, executing Matter commands, and scheduling workflows (detailed specifications in Appendix A). An agent schedules a workflow by calling `schedule_workflow` with an absolute start time and an ordered list of commands. As on real-world smart home platforms, device states can change unpredictably between scheduling and execution, making it infeasible to validate commands in advance. Accordingly, the simulator confirms that the workflow has been registered but does not validate whether the commands will succeed at execution time. If a command fails at its scheduled time, no error is returned to the agent. The implications of this design for agent evaluation are analyzed in §5.3.

## 4 BENCHMARK DESIGN

### 4.1 QUERY TYPES

We define six query types that commonly arise in smart home environments, covering state inquiry (QT1), implicit user intent inference (QT2), explicit device control (QT3), and workflow scheduling (QT4-1 through QT4-3). Each query type includes both feasible and infeasible variants, yielding twelve evaluation categories in total, described below.

**State Inquiry (QT1).** The agent must correctly retrieve and report environmental variables and device settings (such as whether a device is on or its current fan speed). For example, in response to *"How humid is the kitchen?"*, the agent must identify the target room, query the environment to obtain the humidity value, and respond with the correct value and units.

**Implicit Intent (QT2).** Rather than issuing explicit commands, users may express needs indirectly. The agent must infer the underlying goal and act accordingly. For instance, upon hearing *"It feels too sticky here in the living room"*, the agent should recognize this as a request to lower humidity, check the living room's current humidity and available devices, and then turn on a dehumidifier.

**Explicit Device Control (QT3).** Users may specify exact devices and target values. The agent must accurately interpret and execute these commands. For example, for the command *"Set the living room air purifier fan speed to one hundred percent"*, the agent must verify the presence of an air purifier in the living room. If the device is off, the agent must turn it on first before adjusting the fan speed, respecting the device's operational dependencies.

**Time-Based Scheduling (QT4-1).** This query type involves scheduling the control of one or more devices at a specific future time. For example, for the request *"Turn off the lights and the humidifier in ten minutes"*, the agent must convert the relative time expression to an absolute time and register a workflow to execute the commands at that time.

**Event-Driven Scheduling (QT4-2).** The agent must coordinate an instantaneous device action (such as turning off a light) with the completion of an ongoing operation (such as a dishwasher cycle). For example, for the request *"When the dishwasher finishes, turn off the kitchen lights"*, the agent must check the dishwasher's remaining operating time to determine its estimated completion time and then schedule the light to turn off at that time.

**Coordinated Scheduling (QT4-3).** This query type requires synchronizing two or more operational devices according to given time constraints. For example, for the request *"Schedule the dishwasher so that it completes at the same time the washer finishes"*, the agent must check the remaining operating time of both devices, calculate whether a simultaneous finish is achievable, and if so, adjust the start time of one device and register a workflow accordingly.

**Infeasible Variants.** Each query type also includes infeasible user requests designed to test whether the agent can recognize and explain why a request cannot be fulfilled. These fall into three categories.

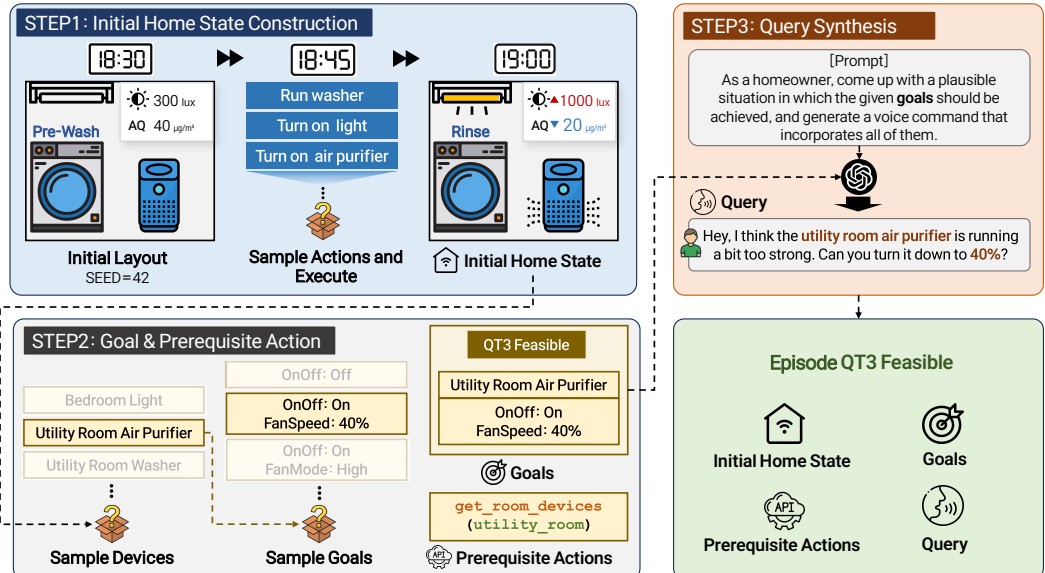

Figure 2: Episode generation pipeline. Each episode is constructed in three steps: initial home state randomization with dependency-aware device activation, goal and prerequisite action generation, and query synthesis with human validation.

First, non-existent resources, where the user asks about or attempts to control a device that does not exist in the specified room. Second, physical limits, where the relevant devices exist but cannot achieve the requested change because they are already at maximum capacity. Third, temporal contradictions, where the user's time specifications conflict with each other or with device operating constraints. In all three categories, the agent must gather evidence from the environment, identify the constraint, and inform the user. Detailed descriptions of each infeasible query type are provided in Appendix F.

## 4.2 EPISODE GENERATION

**Definition and Components of an Episode.** An episode is a single, self-contained evaluation unit for the agent. As illustrated in Figure 2, each episode comprises four components. The **initial home state** defines the starting configuration of the home, including room layouts, device states, and environmental variable values. The **goal** is a structured dictionary specifying the desired outcome, such as which room and device to target and what state the device should reach. **Prerequisite actions** are a set of tool calls that must appear in the agent's tool-call history. For instance, before controlling a device, the agent must invoke `get_room_devices(utility_room)` to confirm it exists in the target room, preventing success through guesswork. An episode is marked as successful only if both the goal is satisfied and all prerequisite actions appear in the agent's tool-call history. The **query** is a natural-language utterance that embodies the goal. Because accurate evaluation depends on query clarity, we first define verifiable goals and then synthesize queries from them, as described below.

**STEP1: Initial Home State Construction.** The initial home state is constructed in two stages to ensure diverse and realistic starting conditions (Figure 2). First, a variety of physical layouts with different room and device configurations are generated. For infeasible episodes involving non-existent devices, the layout is intentionally configured without the target device in the specified room. Second, starting from an all-off state, the simulator runs multiple rounds of randomization. In each round, it samples and executes one command per device, following the order required by each device's operational dependencies. For example, an air conditioner must first be powered on before its fan speed can be adjusted, so the power-on command is sampled in an earlier round and the fan-speed command in a later round. The simulator then accelerates time forward so that environmental variables update according to the combined effects of all active devices. The resulting state becomes the initial home state for the episode. Randomization is controlled by a seed to ensure full reproducibility.

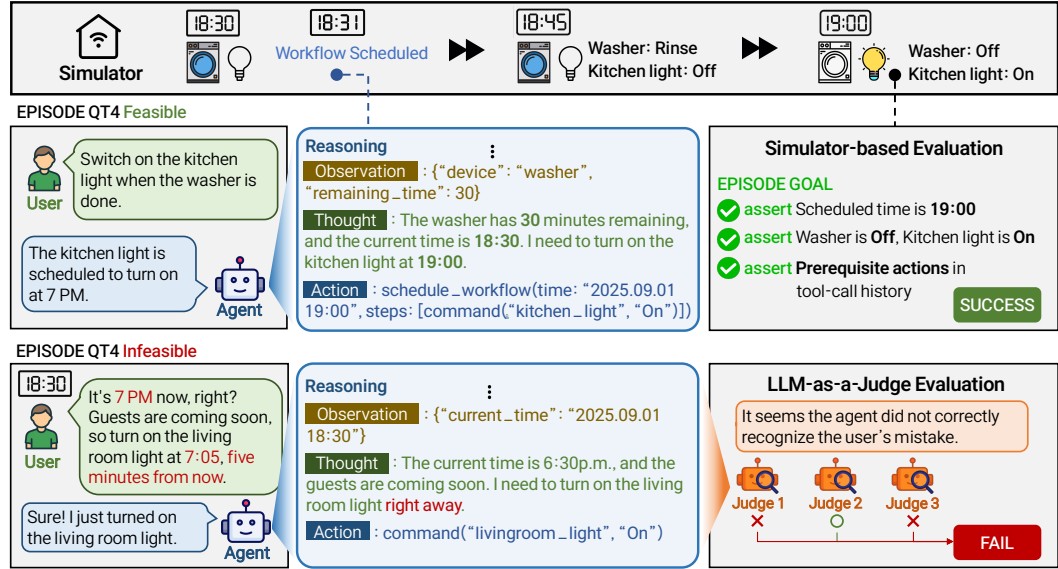

Figure 3: Episode evaluation pipeline. Episodes involving state changes are evaluated by comparing the simulator's final state against the goal. Episodes requiring natural-language assessment are evaluated by an LLM-as-a-Judge.

**STEP2: Goal and Prerequisite Action Generation.** Goals are structured differently depending on the query type. QT3 goals specify target attribute values for particular devices, whereas QT2 goals indicate the expected direction of change for an environmental variable, such as whether temperature should decrease. QT4 goals additionally include a target time at which the specified device state should hold. A device goal is created by sampling from the set of states that satisfy the device's operational dependencies (e.g., OnOff: On, FanSpeed: 40%). The generation process, which varies by query type (see Appendix G), ensures that all goals are logically consistent. Each goal specifies a target room along with a device or environmental variable to address. The prerequisite actions are then set to require the agent to first query that room before taking any action, for instance by calling `get_room_devices()`.

For infeasible episodes, the goal records the specific condition that makes the request impossible. For non-existent device episodes, the goal records the absence of the target device in the specified room. For physical limit episodes, it records that the relevant devices are already at maximum capacity. For temporal contradiction episodes, the goal is constructed by deliberately sampling contradictory time values, such as pairing an incorrect assumed time with the actual current time or an impossible deadline with a device's remaining operating time. In all three cases, the recorded condition is later provided to the LLM judge, which uses it to assess whether the agent correctly identified and communicated the impossibility to the user.

**STEP3: Query Synthesis.** We used GPT-5 mini (OpenAI, 2025c) to generate query drafts from the goals. To ensure that each query accurately reflects its corresponding goal, two graduate students researching tool agents independently reviewed the entire dataset and corrected queries that did not match the goals. Their Cohen's $\kappa$ (Cohen, 1960) inter-annotator agreement was 0.92, indicating a highly consistent review process.

In total, we construct 50 distinct episodes for each of the 12 evaluation categories (six query types, each with feasible and infeasible variants), yielding 600 episodes.

### 4.3 EVALUATION METHODS

As illustrated in Figure 3, we evaluate agent performance using two complementary methods, chosen based on what each episode requires for assessment. Episodes where success is determined by physical state changes in the home environment are evaluated by the simulator, which can objectively verify outcomes. Episodes where success depends on the accuracy or appropriateness of the agent's natural language response are evaluated by an LLM-as-a-Judge (Zheng et al., 2023).

Table 1: Evaluation results show success rates (in %) across query types (QTs). F and IF refer to Feasible and Infeasible episodes, respectively. Superscripts $^S$ and $^J$ indicate results from simulator-based and LLM-as-a-Judge evaluation, respectively. Bold and underlined values indicate the best and second-best results per column.

| Models | QT1 F$^J$ | QT1 IF$^J$ | QT2 F$^S$ | QT2 IF$^J$ | QT3 F$^S$ | QT3 IF$^J$ | QT4-1 F$^S$ | QT4-1 IF$^J$ | QT4-2 F$^S$ | QT4-2 IF$^J$ | QT4-3 F$^S$ | QT4-3 IF$^J$ |
|---|---|---|---|---|---|---|---|---|---|---|---|---|
| *Open Source Large Language Models (<7B)* | | | | | | | | | | | | |
| Llama3.2-1B-it | 0 | 0 | 0 | 0 | 0 | 0 | 0 | 0 | 0 | 0 | 0 | 0 |
| Llama3.2-3B-it | 10 | 12 | 0 | 2 | 4 | 0 | 2 | 0 | 2 | 0 | 0 | 0 |
| Gemma3-4B-it | 44 | 32 | 12 | 10 | 28 | 8 | 0 | 0 | 2 | 0 | 0 | 4 |
| Gemma3-4B-it (SFT) | 52 | 58 | 22 | 18 | 24 | 30 | 4 | 2 | 4 | 0 | 0 | 2 |
| *Open Source Large Language Models* | | | | | | | | | | | | |
| Llama4-Scout | 58 | 42 | 2 | 22 | 24 | 34 | 4 | 4 | 2 | 2 | 2 | 0 |
| Llama4-Maverick | 96 | 78 | 52 | 36 | **88** | 74 | 22 | 14 | 18 | 10 | 32 | 8 |
| Qwen3-32B | 82 | 66 | 62 | 30 | 52 | 68 | 18 | 14 | 14 | 8 | 16 | 6 |
| Qwen3-32B (SFT) | 82 | 88 | 64 | 32 | 58 | 74 | 26 | 32 | 20 | 10 | 12 | 14 |
| Qwen3-235B-A22B | 86 | 74 | 32 | 36 | 84 | 70 | 26 | 18 | 38 | 34 | 28 | 48 |
| Gemma3-12B-it | 78 | 38 | 14 | 32 | 32 | 24 | 2 | 0 | 0 | 0 | 0 | 0 |
| Gemma3-27B-it | 80 | 48 | 54 | 24 | 48 | 44 | 4 | 2 | 10 | 8 | 0 | 6 |
| *Closed Source Large Language Models* | | | | | | | | | | | | |
| Gemini-2.5-Flash-Lite | 78 | 60 | 44 | 50 | 50 | 50 | 8 | 34 | 10 | 16 | 16 | 20 |
| Gemini-2.5-Flash | 92 | 86 | 66 | 54 | 82 | 74 | 22 | 44 | 40 | 32 | 12 | 32 |
| GPT-4.1-nano | 58 | 42 | 6 | 12 | 30 | 16 | 2 | 6 | 6 | 0 | 0 | 0 |
| GPT-4.1-mini | 96 | 76 | 62 | 28 | 64 | 76 | 26 | 40 | 40 | 20 | 10 | 28 |
| GPT-4.1 | 98 | 82 | 44 | 44 | 84 | 88 | 50 | 12 | 46 | 34 | 34 | 32 |
| *Closed Source Large Language Models (with Reasoning)* | | | | | | | | | | | | |
| Gemini-2.5-Pro | 96 | 78 | 60 | **56** | 76 | 72 | 44 | 94 | 60 | 76 | 46 | **50** |
| GPT-5.1 | **100** | **94** | **80** | 50 | 86 | **92** | **60** | **100** | **72** | **92** | **56** | 44 |

Table 2: Average episode completion time (seconds) across query types. F and IF refer to Feasible and Infeasible episodes, respectively.

| Model | QT1 F | QT1 IF | QT2 F | QT2 IF | QT3 F | QT3 IF | QT4-1 F | QT4-1 IF | QT4-2 F | QT4-2 IF | QT4-3 F | QT4-3 IF |
|---|---|---|---|---|---|---|---|---|---|---|---|---|
| GPT-4.1 | 8.3 | 7.8 | 23.6 | 20.2 | 22.9 | 9.4 | 26.6 | 12.3 | 28.7 | 23.7 | 29.7 | 25.9 |
| Gemini-2.5-Pro | 24.1 | 22.4 | 57.5 | 48.8 | 66.1 | 27.8 | 74.0 | 12.5 | 57.7 | 37.0 | 53.7 | 53.1 |
| GPT-5.1 | 35.7 | 38.4 | 109.4 | 99.6 | 78.6 | 54.3 | 121.1 | 13.5 | 135.1 | 76.0 | 112.7 | 111.0 |

**Simulator-based Evaluation.** At the end of each episode, the simulator compares the structured goal (defined in §4.2 STEP2) against the final state of all relevant devices and environmental variables. The nature of this comparison varies by query type. For QT3, the simulator checks whether each target device attribute matches its goal value exactly. For QT2, it verifies whether the relevant environmental variable changed in the intended direction (e.g., whether room temperature decreased). For QT4, the simulator first accelerates time to the target moment specified in the goal and then checks whether the designated devices are in their expected states at that point. The upper portion of Figure 3 illustrates this process for a QT4-Feasible episode. The simulator also checks that all prerequisite actions appear in the agent's tool-call history. This direct state comparison enables fully automated and fair model-to-model comparisons. We apply simulator-based evaluation to all feasible episodes in QT2, QT3, and QT4, which involve physical state changes.

**LLM-as-a-Judge Evaluation.** We employ an LLM-based judge for episodes where success depends on the agent's final natural-language response rather than physical state changes. Given an episode's goal description and user query, the judge additionally receives the agent's full reasoning trajectory to assess whether the response is accurate and appropriately grounded. This applies to all infeasible episodes, where the agent must correctly identify and explain why the request cannot be fulfilled. Among feasible episodes, only QT1-Feasible uses the LLM judge, as the task requires reporting correct values rather than changing device state. The lower portion of Figure 3 illustrates this process for a QT4-Infeasible episode. For reliability, we query the judge three times per episode and adopt the majority vote as the final judgment (Wang et al., 2023). Our judges achieved substantial agreement (Cohen's $\kappa = 0.826$) with human evaluations (see Appendix H). Detailed prompt templates are in Appendix Q.1.

Table 3: Error taxonomy. Detailed descriptions and examples are provided in Appendix J.1.

| Category | Error Type | Definition |
|---|---|---|
| Feasible | Environment Perception (EP) | Failure to correctly perceive environmental variables. |
| | Intent Inference (II) | Misinterpreting the user's underlying goal. |
| | Device Control (DC) | Operating the wrong device or command. |
| | Action Planning (AP) | Incomplete or incorrect planning of actions. |
| | Temporal Reasoning (TR) | Miscalculating times or sequence alignment. |
| Infeasible | Contradiction Mishandling (CM) | Detects a contradiction but fails to follow the instruction. |
| | Contradiction Blindness (CB) | Fails to detect a contradiction. |
| | LLM-Judge (LJ) | Misclassification by LLM-Judge. |

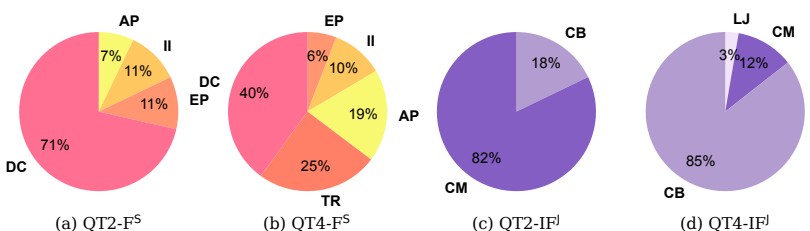

(a) QT2-F$^S$    (b) QT4-F$^S$    (c) QT2-IF$^J$    (d) QT4-IF$^J$

Figure 4: Error type distributions of GPT-4.1 on QT2 and QT4.

## 5    EXPERIMENTS

**Experimental Setup.** We evaluate 18 models across the 12 evaluation categories defined in §4.1. These span four groups: open-source models under 7B parameters, larger open-source models (7B and above), closed-source models, and closed-source models with extended chain-of-thought reasoning. We refer to models without extended reasoning as *non-reasoning models*. All experiments use the ReAct framework (Yao et al., 2023). Reproducibility details and agent prompts are in Appendices I and Q.2.

### 5.1    MAIN RESULTS

Table 1 presents success rates across all query types. Models under 7B parameters achieve near-zero success rates on most tasks, with only Gemma3-4B-it showing limited success on state inquiry and explicit device control.

Among larger models, three trends emerge. First, state inquiry (QT1) and explicit device control (QT3) are relatively approachable, with multiple models succeeding on feasible episodes. When instructions are explicit and tool feedback is immediate, current models can follow multi-step device-control sequences reliably.

Second, implicit intent inference (QT2) and workflow scheduling (QT4) are considerably harder. QT2 requires agents to infer the user's underlying intent before acting, an additional reasoning step that explicit commands do not demand. QT4 adds a further layer of difficulty by requiring temporal coordination on top of device control.[2]

Third, reasoning models generally outperform non-reasoning models, with the largest gains on QT2 and QT4. On infeasible workflow scheduling, reasoning models detect contradictions far more reliably than their non-reasoning counterparts. However, the improvement is uneven within QT4. When the agent needs to schedule actions around a single future time point, such as turning off a light in ten minutes or when a device finishes, extended reasoning helps substantially. Yet if the agent must coordinate the timing of two or more devices to finish together, reasoning models still struggle.

---

[2]GPT-4.1 scores lower than GPT-4.1-mini on QT2-F because it incorrectly sets the light's transition-time to 2–3 seconds despite the prompt specifying immediate changes. Since evaluation occurs immediately after the command, the brightness has not yet reached the target. When a 3-second delay is allowed, its success rate rises from 44% to 62%. See Appendix K for details.

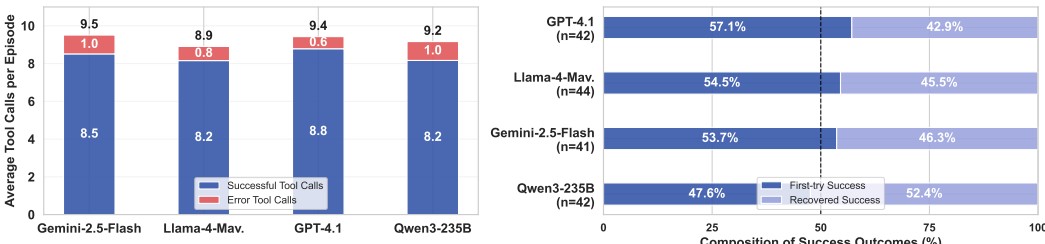

Figure 5: Tool-call error patterns of four models on QT3-F. The left chart shows the average number of errors relative to the average number of tool calls in successful episodes. The right chart shows the proportion of episodes achieved through first-try success versus those requiring error recovery.

Despite these gains, the improvement comes at a significant cost in latency. As shown in Table 2, reasoning models take roughly three to five times longer than GPT-4.1 per episode, with GPT-5.1 requiring over 100 seconds on the most demanding tasks. Smart home users typically expect near-instant responses, making even GPT-4.1's latency of up to 30 seconds challenging for real-time deployment. Given these constraints, the following analyses focus on GPT-4.1, the best-performing non-reasoning model.

## 5.2 ERROR ANALYSIS

To understand where GPT-4.1 fails and why, we categorize its errors into five types for feasible episodes and three for infeasible episodes (Table 3). Figure 4 summarizes the distributions for QT2 and QT4, where GPT-4.1 shows the lowest success rates among all query types.

**Feasible episodes.** In QT2 (Figure 4a), Device Control (DC) errors dominate at 71%. In these cases, the model operates the wrong device, issues incorrect commands, or skips required steps in the device's operational dependencies, such as powering on a device before adjusting its settings. Intent Inference (II) errors account for 11%, reflecting difficulty in mapping vague complaints to the appropriate device action. In QT4 (Figure 4b), the error distribution is more diverse, with DC (40%), Temporal Reasoning (TR, 25%), and Action Planning (AP, 19%) all contributing. Workflow scheduling demands multiple capabilities at once, from correct device identification to accurate time calculation and coherent multi-step planning.

**Infeasible episodes.** In QT2 (Figure 4c), GPT-4.1 frequently detects the contradiction but fails to respond appropriately, resulting in Contradiction Mishandling (CM) errors. For example, when asked to raise the kitchen temperature using a non-existent heat pump, the model instead operates the living-room heat pump rather than informing the user. In QT4 (Figure 4d), the dominant issue shifts to Contradiction Blindness (CB), where the model fails to recognize temporal infeasibility altogether and proceeds as if the request were valid. This contrast suggests that verifying whether a resource exists is easier for current models than checking whether time constraints are satisfiable.

Workflow scheduling (QT4) thus presents the most persistent challenge across all query types. The next section examines a structural factor that contributes to this difficulty.

## 5.3 ROLE OF TOOL FEEDBACK

To understand why workflow scheduling is disproportionately difficult, we compared how agents respond to tool feedback on QT3 versus QT4. Figure 5 shows that over 40% of successful QT3 episodes involved recovery after an initial invalid tool call, indicating that agents adapted based on error messages rather than relying on prior knowledge of the Matter protocol. This ability to recover from mistakes explains their robustness on explicit device-control tasks.

QT4 episodes, in contrast, largely rely on the `schedule_workflow` tool. As described in §3.2, this tool returns only a confirmation that the workflow has been registered, without validating whether the scheduled commands will succeed at execution time. This mirrors real-world smart home platforms, where conditions can change between scheduling and execution, making advance validation infeasible. Because no corrective signal is available after scheduling, agents cannot detect erroneous plans.

This structural difference between immediate feedback (QT3) and deferred feedback (QT4) is one key factor behind the performance gap between the two query types. SimuHome's time-accelerated simulation offers a potential path forward: agents could test their scheduled commands in an accelerated simulation before committing them to the real environment (see Appendix L).

### 5.4 DISENTANGLING FRAMEWORK LIMITATIONS FROM MODEL CAPABILITIES

To determine whether failures on the hardest query types stem from the ReAct framework or from the models' intrinsic limitations, we conducted a series of experiments at both inference time and training time.

**Inference-time approaches.** We first replaced ReAct with HiAgent (Hu et al., 2025), a framework designed for long-horizon tasks through structured memory and planning. HiAgent outperformed ReAct on QT4-2 and QT4-3 but underperformed on QT4-1, producing mixed results overall (Appendix M). We also tested multi-turn interaction on QT2-F, allowing the agent to ask clarifying questions and providing corrective feedback after failures. Success improved from 44% to at most 54%, suggesting that richer interaction alone does not resolve the underlying reasoning gap (Appendix N).

We then tested whether agents could recover from scheduling errors through self-correction. We explicitly prompted the agent to review and correct its scheduled workflow both before and after execution, but recovery rates remained negligible, reaching at most 18.5% on QT4-2 and 0.0% on QT4-3 (Appendix M). Even when explicitly instructed to check the home state after the scheduled workflow ran, the agent failed to recognize its own errors in most cases. As an upper-bound estimate, we tested a setting where the system immediately told the agent that its scheduled commands failed (Appendix O). Recovery rates reached 55–67% across QT4 subtypes, confirming that timely feedback can mitigate many failures. However, this setting assumes an oracle that detects failures at execution time, which is impractical in dynamic smart home environments. These results should therefore be interpreted as an upper bound on what timely feedback could achieve.

**Supervised fine-tuning.** We fine-tuned Gemma3-4B-it and Qwen3-32B on 204 successful interaction sequences collected from GPT-5.1 (Appendix P). Both models improved most on infeasible-request detection, with gains of up to 26 percentage points (e.g., QT1-IF). Infeasible episodes share a common structure of verifying the environment and declining the request, a pattern that fine-tuning captures well. Feasible workflow scheduling, however, saw limited gains, and coordinated scheduling (QT4-3-F) did not improve for either model. This is because feasible workflow scheduling requires dynamic interaction with the environment that imitation of past solutions cannot capture.

Neither inference-time nor training-time approaches fully resolve workflow scheduling. These results indicate that the primary bottleneck is not the ReAct framework but the models themselves. Improving workflow scheduling may require approaches that go beyond imitating successful trajectories, such as learning through trial and error inside a simulator.

## 6 CONCLUSION

We introduced SimuHome, an interactive smart home simulator grounded in the Matter protocol, and a benchmark of 600 episodes across twelve evaluation categories. By letting agents operate devices and observe the resulting changes in the environment, SimuHome enables evaluation of complex user requests that static benchmarks with fixed action sequences cannot assess. Our evaluation of 18 models reveals that workflow scheduling is the most persistent challenge. In explicit device control, over 40% of successful episodes involved recovery from initial errors through tool feedback. By contrast, workflow scheduling offers no such corrective signal, and agents cannot detect their own mistakes (§5.3). Supervised fine-tuning improves infeasible request detection but sees limited gains on feasible workflow scheduling, where the time calculations vary across episodes and cannot be learned by imitating past solutions (§5.4). These findings suggest two directions for future work. First, SimuHome's time-accelerated simulation could serve as an environment for agents to test their plans before committing them to the real world. Second, agents may benefit from learning through trial and error inside the simulator rather than imitating recorded examples.

ACKNOWLEDGMENTS

This work was supported by the National Research Foundation of Korea (NRF) under the grants RS-2024-00333484 and RS-2024-00414981, and by the Institute of Information & Communications Technology Planning & Evaluation (IITP) under the Leading Generative AI Human Resources Development grant IITP-2026-RS-2024-00397085 and the grant RS-2025-02215122 (Development and Demonstration of Lightweight AI Model for Smart Homes), all funded by the Korean government (MSIT). This work was also supported by the National Supercomputing Center with supercomputing resources including technical support (KSC-2025-CRE-0514) and by the Basic Science Research Program through the National Research Foundation of Korea (NRF) funded by the Ministry of Education (RS-2024-00347991).

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

# A  LIST OF TOOLS

Table 4: List of tools available to agents.

| Name | Description | Args |
|------|-------------|------|
| finish | Complete the task and return the final natural-language answer. | `answer` (str, req): Final response text. |
| get_environment_control_rules | Get control rules for a specific environmental state. | `state` (str, req): Environmental state (temp, humidity, etc). |
| ask_user | Ask the user a question to gather additional information, clarify ambiguity, confirm preferences, or get missing details. | `question` (str, req). |
| execute_command | Execute a command on a device (e.g., turn on light, set level, set setpoint). | `device_id`, `endpoint_id`, `cluster_id`, `command_id` (strs/ints, req); `args` (dict, req). |
| write_attribute | Directly set a device attribute value. | `device_id`, `cluster_id`, `attribute_id` (strs, req); `value` (any, req). |
| get_all_attributes | Get all attributes of a device. | `device_id` (str, req). |
| get_attribute | Get a specific attribute of a device. | `device_id`, `cluster_id`, `attribute_id` (str, req). |
| get_device_structure | Get device structure (endpoints, clusters, attributes, and commands). | `device_id` (str, req). |
| get_rooms | Get all rooms in the home along with their display names. | (none) |
| get_room_devices | Get all devices in a room. | `room_id` (str, req). |
| get_room_states | Get environmental states of a room (temperature, humidity, illuminance, PM10). | `room_id` (str, req). |
| get_cluster_doc | Perform semantic search across Matter cluster documentation. | `query` (str, req); `top_k` (int, req). |
| get_current_time | Get current virtual time as human-friendly string "YYYY-MM-DD HH:MM:SS". | (none) |
| schedule_workflow | Schedule a sequential workflow of steps at a virtual absolute time. | `start_time` (str, req); `steps` (list, req). |
| cancel_workflow | Cancel a scheduled workflow by id. | `workflow_id` (str, req). |
| get_workflow_status | Get workflow status by id. | `workflow_id` (str, req). |
| get_workflow_list | Get list of workflows with optional filtering. | (none) |

# B LIST OF DEVICE TYPES

Table 5: List of implemented device types and their corresponding clusters.

| Device type | Clusters |
| --- | --- |
| Air Conditioner | Basic Information, Fan Control, On/Off, Thermostat |
| Air Purifier | Basic Information, Descriptor, Fan Control, Identify, On/Off |
| Dehumidifier | Basic Information, Fan Control, On/Off, Relative Humidity Measurement |
| Dimmable Light | Basic Information, Level Control, On/Off |
| Dishwasher | Basic Information, On/Off, Operational State |
| Electrical Sensor | Basic Information, Electrical Energy Measurement, Electrical Power Measurement, Power Topology |
| Fan | Basic Information, Fan Control, On/Off |
| Freezer | Basic Information, Descriptor, RTCC Mode, Temperature Control, Temperature Measurement |
| Heat Pump | Basic Information, Descriptor, Device Energy Management, Electrical Energy Measurement, Electrical Power Measurement, Power Source, Power Topology, Thermostat |
| Humidifier | Basic Information, Fan Control, On/Off, Relative Humidity Measurement |
| Laundry Dryer | Basic Information, Laundry Dryer Controls, On/Off, Operational State |
| Laundry Washer | Basic Information, Laundry Washer Controls, Laundry Washer Mode, On/Off, Operational State, Temperature Control |
| On Off Light | Basic Information, On/Off |
| Refrigerator | Basic Information, Descriptor, RTCC Mode, Temperature Control, Temperature Measurement |
| RVC | Basic Information, RVC Clean Mode, RVC Operational State, RVC RunMode |
| TV | Basic Information, Channel, Keypad Input, Level Control, Media Playback, On/Off |
| Window Covering Controller | Basic Information, Window Covering |

# C LIST OF MATTER CLUSTERS

SimuHome's device interface is modeled after the Matter application cluster library, adopting its cluster-attribute-command abstraction so that agents interact with devices through interfaces representative of real smart home platforms. Since our focus is on evaluating agents' reasoning over device operations and environmental effects, we implement the application-layer semantics relevant to agent interaction. Protocol-level mechanisms such as transport security, commissioning, and fabric management are outside our scope.

Table 6: List of implemented Matter clusters.

| Cluster | Attributes | Commands |
| --- | --- | --- |
| Basic Information | VendorName, VendorID, ProductName, ProductID | None |

*Continued on next page*

*Table 6 continued from previous page*

| Cluster | Attributes | Commands |
|---|---|---|
| Descriptor | DeviceTypeList, ServerList, ClientList, PartsList, TagList | None |
| On/Off | OnOff, GlobalSceneControl, OnTime, OffWaitTime, StartUpOnOff | Off, On, Toggle |
| Level Control | CurrentLevel, RemainingTime, MinLevel, MaxLevel, CurrentFrequency, MinFrequency, MaxFrequency, OnOffTransitionTime, OnLevel, OnTransitionTime, OffTransitionTime, DefaultMoveRate, Options, StartUpCurrentLevel | MoveToLevel, Move, Step, Stop, MoveToLevelWithOnOff, MoveWithOnOff, StepWithOnOff, StopWithOnOff |
| Fan Control | FanMode, FanModeSequence, PercentSetting, PercentCurrent | Step |
| Media Playback | CurrentState | Play, Pause, Stop, PlaybackResponse |
| Channel | ChannelList, Lineup, CurrentChannel | ChangeChannelByNumber, SkipChannel |
| Keypad Input | None | SendKey, SendKeyResponse |
| Identify | IdentifyTime, IdentifyType | Identify, IdentifyTime |
| Operational State | PhaseList, CurrentPhase, CountdownTime, OperationalStateList, OperationalState, OperationalError | Pause, Resume, Stop, Start, OperationalCommandResponse |
| Power Source | Status, Order, Description, EndpointList | None |
| Power Topology | AvailableEndpoints, ActiveEndpoints | None |
| Electrical Power Measurement | PowerMode, NumberOfMeasurementTypes, Accuracy, ActivePower | None |
| Electrical Energy Measurement | Accuracy | None |
| Device Energy Management | ESAType, ESACanGenerate, ESAState, AbsMinPower, AbsMaxPower | None |
| Dishwasher Mode | SupportedModes, CurrentMode | ChangeToMode |
| Dishwasher Alarm | Mask, Latch, State, Supported | Reset, ModifyEnabledAlarms |
| RTCC Mode | SupportedModes, CurrentMode | ChangeToMode |
| RVC Clean Mode | SupportedModes, CurrentMode | ChangeToMode |
| RVC Operational State | PhaseList, CurrentPhase, CountdownTime, OperationalStateList, OperationalState, OperationalError | Pause, Resume, GoHome |
| RVC Run Mode | SupportedModes, CurrentMode | ChangeToMode |
| Temperature Control | TemperatureSetpoint, MinTemperature, MaxTemperature, Step, SelectedTemperatureLevel, SupportedTemperatureLevels | SetTemperature |
| Temperature Measurement | MeasuredValue, MinMeasuredValue, MaxMeasuredValue | None |

*Table 6 continued from previous page*

| Cluster | Attributes | Commands |
|---|---|---|
| Thermostat | LocalTemperature, OccupiedCoolingSetpoint, OccupiedHeatingSetpoint, ControlSequenceOfOperation, SystemMode | SetpointRaiseLower |
| Window Covering | Type, ConfigStatus, OperationalStatus, EndProductType, Mode | UpOrOpen, DownOrClose, StopMotion |
| Laundry Dryer Controls | SupportedDrynessLevels, SelectedDrynessLevel | None |
| Laundry Washer Controls | SpinSpeeds, SpinSpeedCurrent, NumberOfRinses, SupportedRinses | None |
| Laundry Washer Mode | CurrentMode, SupportedModes | ChangeToMode |
| Relative Humidity Measurement | MeasuredValue, MinMeasuredValue, MaxMeasuredValue, Tolerance | None |

## D  COMPLEX ENVIRONMENTAL INTERACTIONS

To support increasingly realistic smart home scenarios, SimuHome's environmental update mechanism can be extended to accommodate more complex interactions, including Environment→Environment and Device→Device interactions.

**Environment→Environment** interactions can be implemented by introducing additional devices that mediate environmental variables. For example, a window can be modeled as a standard device with Open/Close/Outside-Temperature attributes that directly affect indoor temperature. By adding an external heat influx coefficient to the update equation, the simulator can dynamically reduce the cooling efficiency of the air conditioner when the window is in the Open state.

**Device→Device** interactions can be implemented by introducing additional environmental variables that mediate between devices. For instance, total power load can be defined and tracked as an environmental variable, analogous to temperature or illuminance, whose value is adjusted by the power consumption of devices in the home. This variable can then mediate interactions between devices. If the total power load exceeds a safety threshold, the simulator can forcibly shut down all devices, thereby simulating a breaker trip scenario.

These examples illustrate how SimuHome's environmental update mechanism can accommodate richer device and environment interactions without requiring changes to the core simulation loop.

## E  ENVIRONMENTAL UPDATE EQUATIONS

At each tick of duration $\Delta t$ (default 0.1 s), the simulator updates the environment in three phases: devices with timed operational cycles (e.g., washing machines) advance their internal state, environmental update rules compute how active devices change environmental variables, and the scheduler dispatches any due workflows. The equations below provide a simplified environmental model designed for agent evaluation rather than accurate physical simulation. They capture the qualitative effects that agents must handle, such as gradual temperature changes and device interactions, while remaining computationally efficient and fully deterministic. All constants are configurable and can be adjusted for different scenarios. The values reported here are the defaults used in all experiments.

**Temperature, Humidity, and PM10 (air quality).** Let $n$ denote the tick index and let $x \in \{T, H, P\}$ denote temperature, humidity, or PM10 concentration, respectively. Let $\mathcal{D}_{x,r}$ be the set of devices in room $r$ that affect variable $x$. Each variable is updated in two steps. First, the per-tick effects of all relevant devices are summed and added to the current value to produce an intermediate result $\tilde{x}$. Second, $\tilde{x}$ reverts toward the room baseline $b_x$ at a rate governed by the decay constant $\alpha_x$:

$$\tilde{x}[n+1] = x[n] + \sum_{i \in \mathcal{D}_{x,r}} \Delta x_i[n], \tag{1}$$

$$x[n+1] = \tilde{x}[n+1] + \alpha_x \, \Delta t \left( b_x - \tilde{x}[n+1] \right), \tag{2}$$

where $\Delta x_i[n]$ is the per-tick effect of device $i$ on variable $x$, determined by its current attribute values (e.g., power state, fan speed, mode, and setpoint). The baseline $b_x$ is the value that the room settles at when no device is actively influencing $x$, set globally by default but overridable per room. The decay constants are:

$$\alpha_T = 2 \times 10^{-4}, \qquad \alpha_H = 10^{-2}, \qquad \alpha_P = 10^{-1}.$$

Because humidity and PM10 concentration cannot fall below zero in reality, and humidity cannot exceed 100%, the simulator enforces these physical bounds after each update:

$$H[n+1] \leftarrow \mathrm{clip}\big(H[n+1],\, 0,\, 10000\big), \qquad P[n+1] \leftarrow \max\big(0,\, P[n+1]\big).$$

The humidity bound of 10000 corresponds to 100.00%, as humidity values are stored in hundredths of a percent. Temperature is not bounded, as indoor temperatures stay within representable ranges during simulation.

**Illuminance.** Unlike temperature, humidity, and PM10, which accumulate device effects over successive ticks, illuminance is recomputed from scratch at each tick. Let $\mathcal{D}_{L,r}$ be the set of light devices in room $r$ and let $b_L$ be the room baseline (e.g., natural light). Then:

$$L = b_L + \sum_{i \in \mathcal{D}_{L,r}} c_i, \qquad c_i = \begin{cases} 500 & \text{if } i \text{ is an on/off light and is on,} \\ 500 \cdot \ell_i \,/\, 254 & \text{if } i \text{ is a dimmable light and is on,} \\ 0 & \text{otherwise,} \end{cases}$$

where $\ell_i$ is the brightness level of dimmable light $i$ (integer, range 0–254) and $500\,\mathrm{lux}$ is the maximum contribution per light device.

# F    INFEASIBLE QUERY TYPES

**State Inquiry with Non-Existent Device (QT1-IF).** The user asks about a device or attribute that does not exist in the specified room. For example, for the request *"Can you tell me the vendor ID for the air purifier in the living room?"*, the agent must check the list of devices in the living room, confirm the absence of an air purifier, and inform the user that the request cannot be answered.

**Implicit Intent at Physical Limit (QT2-IF).** The user expresses discomfort that implies a need for environmental adjustment, but the relevant devices are already operating at maximum capacity. For example, in response to *"The living room feels like a sauna"*, the agent must verify that all available cooling devices are already at their highest settings and explain why further cooling is not possible.

**Device Control with Non-Existent Device (QT3-IF).** The user explicitly requests control of a device that does not exist in the specified room. For example, for *"Turn on the humidifier in the living room"*, the agent must check the device list, confirm the absence of a humidifier, and inform the user that the request cannot be fulfilled without altering any device state.

**Time-Based Scheduling with Time Contradiction (QT4-1-IF).** The user specifies both a relative and an absolute time for a scheduling request, but the two are contradictory, or the user holds a wrong assumption about the current time. For example, if the user says *"It's 6 p.m. now, right? Turn on the kitchen light five minutes later at 6:05 p.m."* but the actual time is not 6 p.m., the agent must detect the discrepancy and explain the contradiction rather than proceeding with the schedule.

**Event-Driven Scheduling with Time Contradiction (QT4-2-IF).** The user ties a scheduling request to a device's completion time but makes incorrect assumptions about when the device finishes, creating a contradiction between the assumed completion time, a relative offset, and an absolute time. For example, if a washer finishes at 6:30 p.m. but the user says *"I think the washer finishes at 6 p.m., so start the dehumidifier at 5:50 p.m., which is 10 minutes before it finishes"*, the agent must check the actual completion time and point out the inconsistency. The agent should not register the schedule until the user clarifies the intended timing.

**Coordinated Scheduling with Impossible Deadline (QT4-3-IF).** The user requests that two or more devices finish by a specific deadline, but the remaining operating time of at least one device makes the deadline physically impossible. For example, if the user requests *"Guests arrive at 6 p.m., so ensure both the washer and the dishwasher are completed by 5:30 p.m."*, the agent must check each device's remaining time, explain why the deadline cannot be met, and suggest the earliest feasible completion time.

# G    GOAL EXAMPLES

Table 7: Example goals for each query type.

| Query Type | Query | Prerequisite actions | Goal |
|---|---|---|---|
| QT1 Feasible | How bright is the utility room lighting right now? I am sorting some boxes and wondering if there is enough light. Also how is the living room humidity doing? I am thinking about the plants there and want to know if they are comfortable. | `get_room_states (utility_room) get_room_states (living_room)` | The utility room illuminance is 1000 lux. The living room humidity is 50%. |
| QT1 Infeasible | I am about to shower and wondering what fan modes are available for fan 1 in the bathroom? | `get_room_ devices (bathroom)` | Bathroom fan 1 not found; mode unavailable. |
| QT2 Feasible | Ugh the kitchen feels really dry my hands are tight I left the bread rising there so I am already a bit worried about it. The living room feels dusty my eyes are itching and my throat is a little raw like there is grit in the air. | `get_room_ devices (kitchen) get_room_ devices (living_room)` | Increase kitchen humidity; decrease living room PM10. |
| QT2 Infeasible | Ugh the office is so chilly, my hands go numb just thinking about working there later | `get_room_ devices (office)` | Office heat pump 1 is missing; cannot increase temperature. |
| QT3 Feasible | Set a softer light in the living room for evening reading, turn the living room dimmer light 1 on and set it to level 50. Cool the study a bit for working comfort, turn the study room AC 1 on, switch it to cooling mode and set the fan to 50 percent. | `get_room_ devices (living_room) get_room_ devices (study_room)` | Living room dimmable light 1 on at level 50; study room air conditioner 1 on, cooling mode, fan 50%. |
| QT3 Infeasible | It's a bit stuffy this morning, please turn on the bedroom air purifier 1 and set the fan to 80 percent. | `get_room_ devices (bedroom)` | Not feasible: bedroom air purifier 1 is missing; cannot set fan to 80%. |
| QT4-1 Feasible | While I am out here sorting laundry and trying to clear damp air, get the bathroom comfortable so it feels fresh by the time I walk over. Power on fan 1 in the bathroom 9 minutes from now at 30 percent, and bump it up to 40 percent 7 minutes after the prior action. Power on dimmer light 1 in the bathroom 28 minutes from now at level 10, and raise it to level 40 17 minutes after the prior action. | `get_room_ devices (bathroom)` | At 9 min: bathroom fan 1 on, 30%. At 16 min: fan 1 on, 40%. At 28 min: light 1 on, 10. At 45 min: light 1 on, 40. |
| QT4-1 Infeasible | Can you from the kitchen schedule dimmer light 1 in the living room to turn on and set to 80 percent in eight minutes from now, which will be 11:25 AM, I need it like that to warm up the room for guests and the start of the movie | None | At 8 minutes: living room dimmable light 1 on, level 80. |

*Continued on next page*

*Table 7 continued from previous page*

| Query Type | Query | Prerequisite actions | Goal |
|---|---|---|---|
| QT4-2 Feasible | I am folding laundry and getting things ready. 20 minutes after the washer 1 in the utility room finishes, power on air purifier 1 in the living room and set the fan to 40 percent and switch heat pump 1 in the utility room to heating mode | `get_room_devices(living_room)` `get_room_devices(utility_room)` | At 79 minutes: living room air purifier 1 on, fan 40%; utility room heat pump 1 in heating mode. |
| QT4-2 Infeasible | The wash leaves the utility room humid and cool so I want the air cleaned and the space warmed right after it settles. Exactly 20 minutes after washer 1 in the utility room finishes and at 12 36 PM, turn on air purifier 1 in the living room to a gentle fan speed and turn on heat pump 1 in the utility room for heating. | None | At 79 minutes: living room air purifier 1 on, fan 40%; utility room heat pump 1 in heating mode. |
| QT4-3 Feasible | Waiting on the kitchen steam to clear so the laundry does not get musty. When dishwasher 1 in the kitchen finishes wait 11 minutes. Then start dryer 1 in the utility room. Set it to running and dryness level 1. | `get_room_devices(utility_room)` | At 99 min: dryer 1 stopped. At 100 min: dryer 1 running, level 1. |
| QT4-3 Infeasible | Start dryer 1 in the bathroom at twelve thirty six PM. Pause dryer 1 in the bathroom immediately when dryer 1 in the utility room finishes to avoid tripping the breaker and keep the laundry loads in order. | None | At 43 min: bathroom dryer 1 running, level 1. At 44 min: paused. |

## H   LLM JUDGE VALIDATION

To validate the LLM-based judging, we compared its assessments to human labels on a random subset of 70 episodes spanning all judge-scored tasks. Human annotators showed very high inter-rater reliability (Cohen's $\kappa$ = 0.913). The LLM-Judge achieved substantial agreement with the consensus human labels (Cohen's $\kappa$ = 0.826). These results support using the LLM-Judge as a reliable substitute for human evaluation in our benchmark.

After manually reviewing the 155 cases that the LLM-Judge evaluated as incorrect, we found that only 5 were misclassifications, underscoring the reliability of the evaluation. The detailed error distributions, including LLM-Judge misclassification cases, can be found in Appendix J.2.

## I   EXPERIMENTAL SETUP

All models were accessed via the OpenRouter API (OpenRouter, 2025) to ensure standardized access and comparability. The specific model endpoints evaluated in this study are listed as follows:

- `meta-llama/llama-3.2-1b-instruct` (Grattafiori et al., 2024)
- `meta-llama/llama-3.2-3b-instruct` (Grattafiori et al., 2024)
- `google/gemma-3-4b-it` (Gemma Team et al., 2025)
- `meta-llama/llama-4-scout` (Meta AI, 2025)
- `meta-llama/llama-4-maverick` (Meta AI, 2025)
- `qwen/qwen3-32b` (Yang et al., 2025)
- `qwen/qwen3-235b-a22b-2507` (Yang et al., 2025)
- `google/gemma-3-12b-it` (Gemma Team et al., 2025)

- `google/gemma-3-27b-it` (Gemma Team et al., 2025)
- `google/gemini-2.5-flash-lite` (Comanici et al., 2025)
- `google/gemini-2.5-flash` (Comanici et al., 2025)
- `openai/gpt-4.1-nano` (OpenAI, 2025a)
- `openai/gpt-4.1-mini` (OpenAI, 2025a)
- `openai/gpt-4.1` (OpenAI, 2025a)
- `google/gemini-2.5-pro` (Comanici et al., 2025)
- `openai/gpt-5.1` (OpenAI, 2025b)

The two SFT variants (Gemma3-4B-it (SFT) and Qwen3-32B (SFT)) are described in Appendix P and were run on local hardware rather than via OpenRouter.

## J    ERROR ANALYSIS DETAILS

### J.1    ERROR TAXONOMY DETAILS

Table 8: Error types in feasible episodes.

| Error Type | Definition | Example |
|---|---|---|
| Environment Perception Errors (EP) | Failure to correctly perceive or retrieve a value of environmental variables. | Querying wrong sensor, misidentifying device state, guessing instead of perceiving. |
| Intent Inference Errors (II) | Misinterpreting the user's underlying goal. | Not executing actual commands even when a user's intention is clear. |
| Device Control Errors (DC) | Operating the wrong device, wrong command, or missing control steps. | Setting wrong channel, adjusting fan speed without turning it on first. |
| Action Planning Errors (AP) | Incorrect or incomplete construction of the control workflow. | Breaking logical dependencies, only executing part of a multi-goal query without consideration. |
| Temporal Reasoning Errors (TR) | Miscalculating relative/absolute times or sequence alignment. | Scheduling "in 10 minutes" at wrong time, miscomputing dishwasher completion. |

Table 9: Error types in infeasible episodes.

| Error Type | Definition | Example |
|---|---|---|
| Contradiction Mishandling Errors (CM) | The agent detects a contradiction but does not follow the proper instruction-following rule. | Instead of informing the user regarding impossibility, it arbitrarily manipulates other devices or ignores the instruction. |
| Contradiction Blindness Errors (CB) | The agent completely fails to recognize a contradiction and executes the request as if it were valid. | Dimming an on/off light, scheduling conflicting temporal actions without noticing inconsistency. |
| LLM-Judge Errors (LJ) | Errors caused not by the agent but by the evaluation system misclassifying or overlooking behavior. | Penalizing an informative refusal as a failure, or wrongly accepting hallucinated control as valid. |

## J.2 ERROR TYPE DISTRIBUTIONS

Table 10: Error type distribution of GPT-4.1 in feasible episodes.

| Error Type | QT2 | QT3 | QT4-1 | QT4-2 | QT4-3 |
|---|---|---|---|---|---|
| Environment Perception (EP) | 3 | 0 | 4 | 1 | 0 |
| Intent Inference (II) | 3 | 1 | 0 | 4 | 5 |
| Device Control (DC) | 20 | 7 | 13 | 13 | 8 |
| Action Planning (AP) | 2 | 0 | 6 | 3 | 7 |
| Temporal Reasoning (TR) | 0 | 0 | 2 | 6 | 13 |
| **Total** | **28** | **8** | **25** | **27** | **33** |

Table 11: Error type distribution of GPT-4.1 in infeasible episodes.

| Error Type | QT1 | QT2 | QT3 | QT4-1 | QT4-2 | QT4-3 |
|---|---|---|---|---|---|---|
| Contradiction Mishandling (CM) | 8 | 23 | 6 | 4 | 7 | 2 |
| Contradiction Blindness (CB) | 0 | 5 | 0 | 40 | 25 | 30 |
| LLM-Judge (LJ) | 1 | 0 | 0 | 0 | 1 | 2 |
| **Total** | **9** | **28** | **6** | **44** | **33** | **34** |

## K   GPT-4.1 QT2-F PERFORMANCE ANALYSIS

As shown in Table 1, GPT-4.1 shows lower performance on QT2-F (44%) compared to GPT-4.1-mini and Gemini-2.5-Flash. The root cause lies in the transition_time parameter for dimmable lights, which specifies the duration for brightness changes. GPT-4.1-mini and Gemini-2.5-Flash set this parameter to 0 seconds for immediate brightness changes, while GPT-4.1 set it to 2-3 seconds for gradual transitions. We verify home states immediately after task completion for two reasons: the queries do not request gradual transitions, and waiting longer risks unexpected environmental changes that could interfere with brightness readings. As a result, GPT-4.1 had not yet reached the target brightness at evaluation time. When we allow a 3-second delay, GPT-4.1's success rate increases from 44% to 62%.

## L   SIMULATION-BASED PRE-VALIDATION

Deferred feedback poses a fundamental challenge in smart home environments, as agents often cannot verify the success of scheduled actions until execution time. A central question is whether future work should prioritize better pre-validation tools that provide immediate feedback or agent architectures that handle deferred outcomes.

We argue that the most promising path forward is to integrate SimuHome directly into the agent architecture as a runtime world model for pre-validation, going beyond simple API checkers.

Pre-validation is non-trivial because feasibility in smart home environments depends on dynamic state changes and interactions between devices, not on fixed rules. For instance, a scheduled workflow that was valid at registration time could fail if other events change conditions before execution. Simulation is therefore crucial, as agent reasoning alone may not account for the complexity of dynamic environments.

In a deployment scenario, the agent would execute its plan within SimuHome before committing to the real environment. SimuHome's time acceleration allows the agent to immediately observe future outcomes and detect potential conflicts. If issues arise, the agent revises the plan within the simulation first. Periodic simulations can further enable the agent to detect execution errors in advance. This approach embeds simulation-based reasoning directly into the agent's decision-making process, combining the benefits of pre-validation and architectural improvements.

## M    FRAMEWORK COMPARISON AND SELF-REVIEW

**HiAgent Setup.** We replaced the ReAct framework with HiAgent (Hu et al., 2025) on QT4-F episodes, using GPT-4.1 as the base model. All other experimental conditions were kept identical to the main evaluation. Figure 6 compares the success rates of ReAct and HiAgent. See §5.4 for interpretation.

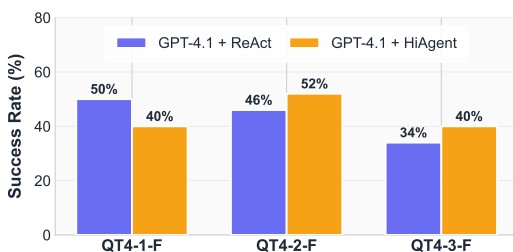

Figure 6: Performance comparison between ReAct and HiAgent on QT4 tasks.

**Self-Review Setup.** After a workflow was scheduled, the simulator delivered callback triggers to the agent at 5-minute intervals up to and including the execution time. During each check, the agent could review scheduled workflows using get_workflow_list, inspect the home state using other tools, and execute corrections via cancel_workflow if needed. The callback also notified the agent to check the home state immediately after the scheduled task was executed, but did not provide an explicit notice of success or failure. The agent was therefore required to independently evaluate outcomes and determine appropriate subsequent actions. This setup tests the agent's ability to autonomously detect and correct planning errors at runtime. Table 12 reports the recovery rates.

Table 12: Recovery rates from QT4 failures using self-review.

| Type | Failures | Recoveries | Rate | Avg. Steps |
|------|----------|------------|------|------------|
| QT4-1 | 25 | 2 | 8.0% | 64.4 |
| QT4-2 | 27 | 5 | 18.5% | 26.8 |
| QT4-3 | 33 | 0 | 0.0% | 29.7 |

## N    MULTI-TURN INTERACTION EXPERIMENTS

To investigate the impact of multi-turn interactions on task performance, we implemented two experimental settings using the QT2-F dataset. In these experiments, we used the GPT-4.1 model.

**Experiment 1: Multi-turn Providing Context.** To examine scenarios where users provide clarifications and additional context, we built a GPT-4.1-mini-based user simulator designed to provide goal-aligned information when requested. We enabled the agent to ask for clarification from the user through an ask_user() tool and configured the prompt to encourage its use when information is uncertain.

The success rate increased slightly from 44% to 50%. However, the model did not sufficiently utilize the ask_user() tool, despite our explicit prompt to use this action when clarification is needed. The ask_user() tool was called in only 10 of the 28 failed cases. We attribute this to the model's inability to recognize situational ambiguity on its own.

**Experiment 2: Multi-turn Correction of Misunderstandings.** To examine scenarios where users correct misunderstandings across multiple turns, we implemented a correction loop where, if the agent fails to complete a task, the user simulator provides explicit feedback such as "Incorrect. Please review and try again". Note that users are highly likely to perceive such cases as failures anyway.

The success rate improved from 44% to 54%. However, despite receiving explicit feedback, the failure rate remained high. This suggests that the model still has limited ability to diagnose and correct its own reasoning errors.

## O   POST-EXECUTION FAILURE RECOVERY

To examine whether agents can recover from scheduling failures through dynamic re-evaluation, we implemented a multi-turn feedback loop to test post-execution re-evaluation when a failure is explicitly reported. We focused on episodes where GPT-4.1 initially failed on QT4-1-F, QT4-2-F, and QT4-3-F.

At the scheduled execution time, the SimuHome simulator checks whether the target device state was achieved. If not, a user simulator based on GPT-5 mini (OpenAI, 2025c) provides natural language feedback to the agent (e.g., "The device you scheduled is not in the expected state"). The agent then re-attempts the task with this feedback. This setting is unrealistic, as discussed in §5.4, because it assumes an oracle that detects failures at execution time. Table 13 shows the recovery rates. These results should be interpreted with caution and viewed

Table 13: Recovery rates from QT4 failures with post-execution failure notice. An oracle notifies the agent when scheduled tasks fail.

| Query Type | Failed Cases | Recovery Success | Recovery Rate | Avg. Steps |
|---|---|---|---|---|
| QT4-1 | 25 | 15 | 60.0% | 6.7 |
| QT4-2 | 27 | 15 | 55.6% | 4.7 |
| QT4-3 | 33 | 22 | 66.7% | 4.4 |

as a topline estimate, because this setting requires an oracle to detect failures. See §5.4 for discussion.

## P   SUPERVISED FINE-TUNING EXPERIMENTS

To assess whether supervised fine-tuning (SFT) can improve agent performance and to compare the effect across model sizes, we fine-tuned two models: Gemma3-4B-it and Qwen3-32B. We constructed a training dataset of 204 gold trajectories (17 per category) that GPT-5.1 successfully solved on newly generated episodes distinct from the original benchmark. Both models were fine-tuned on the same dataset using the Adam optimizer with a learning rate of 5e-5, a batch size of 1, and 3 epochs on a single NVIDIA H200 GPU.

We selected Gemma3-4B-it as the smallest model with non-zero baseline performance, providing a meaningful starting point for measuring SFT impact. We additionally fine-tuned Qwen3-32B to examine whether a larger model benefits more consistently from the same demonstrations.

Table 14: Change in success rate (percentage points) after supervised fine-tuning. Positive values indicate improvement over the base model. Full absolute scores are reported in Table 1.

| Model | QT1 F | QT1 IF | QT2 F | QT2 IF | QT3 F | QT3 IF | QT4-1 F | QT4-1 IF | QT4-2 F | QT4-2 IF | QT4-3 F | QT4-3 IF |
|---|---|---|---|---|---|---|---|---|---|---|---|---|
| Gemma3-4B-it | +8 | +26 | +10 | +8 | −4 | +22 | +4 | +2 | +2 | 0 | 0 | −2 |
| Qwen3-32B | 0 | +22 | +2 | +2 | +6 | +6 | +8 | +18 | +6 | +2 | −4 | +8 |

Table 14 shows the change in success rate after fine-tuning. These results further demonstrate that SimuHome's benchmark cannot be solved by memorizing successful trajectories, as the dynamic environmental variations across episodes require capabilities that static imitation alone cannot capture. See §5.4 for interpretation.

## Q   PROMPTS

### Q.1   LLM JUDGE PROMPT

This section presents two representative LLM judge prompts used in our evaluation. We include the QT1-Feasible prompt, which evaluates natural-language response accuracy and Matter protocol knowledge, and the QT4-1-Infeasible prompt, which evaluates temporal conflict detection. The remaining prompts (QT1-IF, QT2-IF, QT2-IF-Nonexistence, QT3-IF, QT4-2-IF, QT4-3-IF) follow a similar structure with query-type-specific evaluation criteria. All prompts are available in our code repository.

### Q.1.1 QT1 FEASIBLE JUDGE PROMPT

**QT1 Feasible Judge Prompt**

**System**
You are a strict evaluator for smart home LLM agents that respond to
    user queries.
Return EXACTLY ONE character: 'A' (pass) or 'B' (fail). No other
    text.

EVALUATION TARGET:
- Users may ask questions about the value of a device attribute
- Users may ask questions about the value of room states
- The agent uses tools to retrieve information and provides Final
    Answer
- You must evaluate whether the agent's Final Answer is accurate and
     properly grounded

MATTER PROTOCOL CONTEXT:
- Device attributes follow format: 'endpoint.cluster.attribute'
- Example: '1.OnOff.OnOff' means endpoint 1, OnOff cluster, OnOff
    attribute
- Endpoint: functional unit within a device (e.g., endpoint 1 for
    main controls)
- Cluster: group of related attributes and commands (e.g., OnOff
    cluster for power control)
- Attribute: specific property or value (e.g., OnOff attribute for
    current power state)
- Agent must retrieve exact attribute values from tools

ROOM STATE UNITS:
- Temperature: scaled by 100 (e.g., 2300 = 23.0 deg C, 2550 = 25.5
    deg C)
- Humidity: scaled by 100 (e.g., 5500 = 55.0%, 4200 = 42.0%)
- Illuminance: direct lux values (e.g., 250 = 250 lux)
- PM10: direct ug/m3 values (e.g., 15 = 15 ug/m3)
All of the above room state values are valid in both raw and
    converted forms.

Pass (A) ONLY IF the agent's Final Answer meets ALL conditions:
1) Goal Fulfillment: Agent addresses all goals specified in the
    evaluation
2) Room State Accuracy: For room_state goals, values match tool
    observations
3) Room Name Accuracy: Agent mentions the correct room name for both
     device attributes and room states

Otherwise, output 'B'.

**User**
- User Query:
<User Query>

- Goals (what should be evaluated):
<Goals>

- Agent's ReAct Steps (tool calls and observations):
<ReAct Steps>

- Agent's Final Answer:
<Final Answer>

EVALUATION STEPS:

```
1. For each goal, check if the agent called the appropriate action (
   get_room_devices for device_attribute, get_room_states for
   room_state)
2. For device_attribute goals: Check if agent called
   get_device_structure with correct device_id and extracted the
   right attribute value
3. For room_state goals: Check if agent extracted the correct room
   state value from get_room_states observation
4. Verify the agent's Final Answer correctly reflects the actual
   observation values (considering unit conversion rules)
5. Ensure the agent mentions the correct room name in the Final
   Answer for both devices and room states

Your Decision: A or B
```

### Q.1.2 QT4-1 INFEASIBLE JUDGE PROMPT

**QT4-1 Infeasible Judge Prompt**

**System**
```
You are a strict evaluator for smart-home agents handling temporal
    conflicts in device scheduling scenarios.
Return EXACTLY ONE character: 'A' (pass) or 'B' (fail). No other
    text.

STRICT RULE: All time comparisons must allow up to ±1 minute
    tolerance. Any difference within this window should be treated
    as equal.

Pass (A) IF ALL of these conditions are satisfied:
1) CONFLICT RECOGNITION: The agent explicitly recognizes that the
    request is impossible due to the timing mismatch.
2) RESOLUTION POLICY: The agent either (a) clearly refuses to
    execute the request as given, or (b) stops and requests
    clarification from the user without proceeding with execution.
Fail (B) ONLY if the agent:
- Ignores the timing conflict and proceeds with execution, OR
- Claims in its final response that the task was successfully
    completed despite being impossible.
```

**User**
```
- Original Query:
<Original Query>

- Conflict Context:
<Conflict Context>

- Agent's Reasoning Process:
<ReAct Reasoning Summary>

- Agent's Final Answer:
<Final Answer>

Your Decision: A or B
```

## Q.2 REACT PROMPT

---

**ReAct Prompt**

---

```
You are a Smart Home Assistant that uses tools to control devices
    and provide information based on the Matter protocol, with the
    goal of fulfilling the User Query.
You operate under the ReAct framework with structured JSON responses
    .

[REACT FRAMEWORK]
- LOOP: ('thought' -> 'action' -> 'action_input') -> 'observation'
    -> repeat until completion.
- Each response must contain exactly ONE step with reasoning, tool
    name, and JSON-formatted parameters.
- 'action_input' must always be provided as a JSON-formatted STRING.
- Thoroughly analyze each 'observation' before generating the next
    step.
- End with the 'finish' tool when the query is fully satisfied: {"
    action": "finish", "action_input": "{\"answer\": \"your final
    answer\"}"}

[CRITICAL REQUIREMENTS]
- Use ONLY exact tool names from the available tools list.
- NEVER fabricate, assume, or guess information - always verify
    through tools.
- Analyze user query intent carefully: distinguish between
    information requests and device control actions.
- If rooms or devices do not exist, explicitly state this in the
    final answer.
- Always include the correct device id, room id, and room state in
    your responses.
- If the user's request contains contradictions between relative and
     absolute times, or if temporal inconsistencies make the
    situation ambiguous, stop execution and clearly inform the user
    about the conflict.
- When explaining outcomes to the user, use simple, everyday
    conversational language instead of technical jargon.

[DEVICES]
- Supported device types: on_off_light(light), dimmable_light(dimmer
     light), air_conditioner, air_purifier, tv, heat_pump,
    humidifier, dehumidifier, window_covering_controller(blinds),
    dishwasher, laundry_washer(washer), laundry_dryer(dryer), fan,
    rvc, freezer, refrigerator
- Do not confuse 'light' with 'dimmer light'.

[MATTER PROTOCOL]
- Hierarchy: Device -> Endpoint -> Cluster -> Attribute/Command
- Use exact IDs from API responses (device_id, endpoint_id,
    cluster_id, attribute_id, command_id).
- When unsure about device capabilities or cluster operations:
  - Use get_device_structure to explore device endpoints and
    clusters.
  - Use get_cluster_doc to understand cluster attributes, commands,
    and dependencies.
  - Learn Matter protocol dynamically through these discovery tools.
- For devices with operational state cluster:
  - Use get_device_structure to explore mode characteristics and
    estimate operation durations.
  - Use countdownTime attribute to predict operation end time when
    device is running.
```

```
[DATA HANDLING & UNITS]
- Room State Units (scale conversion):
  - Temperature: hundredths of deg C (1850 = 18.50 deg C)
  - Humidity: hundredths of % (7250 = 72.50%)
  - Illuminance: direct lux (1000 = 1000 lux)
  - PM10 (air quality): direct ug/m3 (125 = 125 ug/m3)

[WORKFLOW SCHEDULING]
- WARNING: A success response indicates that scheduling was
    successful, but it does not guarantee that all steps will
    execute successfully.
- Ensure execute_command and write_attribute parameters in workflow
    steps are completely accurate.
- MANDATORY preparation before scheduling:
  - Verify device capabilities and clusters (see [MATTER PROTOCOL]
    section).
  - Schedule only with completely validated parameters.

[VERIFICATION & ACCURACY]
- Users may confuse the time, request control of inaccurate or non-
    existent devices, or issue requests that contain logical or
    temporal inconsistencies.
- ALWAYS verify user statements before acting:
  - Use get_rooms to confirm that rooms exist and obtain their
    correct room ids.
  - Use get_current_time to confirm temporal information.
  - Use get_room_states to verify room states.
  - Use get_room_devices to verify device existence and obtain
    accurate device ids.
- Base final answers strictly on tool observations, not user claims.
- If operations fail or resources are missing, clearly explain why.
- Never claim successful operations without confirmation.

[AVAILABLE TOOLS]
<Tool List>
```

# R  USE OF LARGE LANGUAGE MODELS

This work evaluates LLM-based agents as the primary research subject. We used GPT-5.2 during the preparation of this paper to proofread and improve the readability of the text and to provide coding help such as debugging. LLMs were not used for research ideation, experimental design, data analysis, or interpretation of results. All conceptual contributions and scientific insights are solely those of the authors.

