# OpenReview forum: "SimuHome: A Temporal- and Environment-Aware Benchmark for Smart Home LLM Agents"
_ICLR.cc/2026/Conference — ICLR 2026 Oral_

### Official Review · Reviewer_hnkK · 2025-10-16

**Soundness:** 2
**Presentation:** 3
**Contribution:** 2
**Rating:** 4
**Confidence:** 3

**Summary:**

The paper proposes SimuHome, a simulator and benchmark for evaluating smart home LLM agents under different real-world scenarios. It models 4 environmental variables and 17 device types, and provides 600 episodes across 12 query types. The authors evaluate 11 agents and find that models struggle with latent-intent inference, live-state verification, and temporal scheduling.

**Strengths:**

1: The idea of developing smart home agents capable of dealing with real world problems is interesting and worth exploring.

2: The experiments setting with 600 episodes and 12 different queries is comprehensive.

3: The illustrations in the paper is well-structured and make it easier for the understanding of the paper.

**Weaknesses:**

1: The selection of LLMs is not up-to-date and not optimal. GPT-5 should be taken into consideration. Meanwhile, the authors chose Gemini-2.5-Flash and GPT-4.1 instead of Gemini-2.5 Pro and did not enable the thinking mode for GPT-4.1, which limits the persuasiveness of the claims and experiments.

2: Table 1 is confusing, especially regarding the different superscripts J and S. In QT1, both F and IF are marked with J (for LLM-judge-based evaluation), but for the following queries, the format seems to be inconsistent. Also, in Table 1 for the QT2-F Task, the success rate of GPT-4.1 is 44%, which is far behind GPT-4.1-mini and Gemini-2.5-Flash. I'm curious about the reason for this phenomenon.

**Questions:**

See above

---

> ### Author Response · Authors · 2025-11-26
>
> We deeply appreciate your recognition that developing agents for real-world smart home problems is an interesting and worthwhile direction. We have addressed your questions below.
> ___
> > **W1. The selection of LLMs is not up-to-date and not optimal. GPT-5 should be taken into consideration. Meanwhile, the authors chose Gemini-2.5-Flash and GPT-4.1 instead of Gemini-2.5 Pro and did not enable the thinking mode for GPT-4.1, which limits the persuasiveness of the claims and experiments.**
>
> **A1.** We appreciate your suggestion to include up-to-date models and enable thinking mode. To address this, we have expanded our evaluation to include Gemini2.5-Pro and GPT-5.1. Unfortunately, GPT-4.1 does not support a native thinking mode, so we evaluated the more recent GPT-5.1 with Think Mode.
> In addition, for comprehensiveness, we also further evaluated small models under 7B parameters (Llama3.2-1B/3B-it, Gemma3-4B-it) to explore the feasibility of on-device deployment. The results are shown in the table below. Success rates in the table are presented as percentages.
>
> | Models              | QT1-F | QT1-IF | QT2-F | QT2-IF | QT3-F | QT3-IF | QT4-1-F | QT4-1-IF | QT4-2-F | QT4-2-IF | QT4-3-F | QT4-3-IF |
> |--------------------|-------|--------|-------|--------|-------|--------|---------|----------|---------|----------|---------|----------|
> | Llama3.2-1b-it  | 0.0  | 0.0   | 0.0  | 0.0   | 0.0  | 0.0   | 0.0    | 0.0     | 0.0    | 0.0     | 0.0    | 0.0     |
> | Llama3.2-3b-it  | 10.0 | 12.0  | 0.0  | 2.0   | 4.0  | 0.0   | 2.0    | 0.0     | 2.0    | 0.0     | 0.0    | 0.0     |
> | Gemma3-4b-it      | 44.0 | 32.0  | 12.0 | 10.0  | 28.0 | 8.0   | 0.0    | 0.0     | 2.0    | 0.0     | 0.0    | 4.0     |
> | Gemini2.5-pro     | 96.0 | 78.0  | 60.0 | 56.0  | 76.0 | 72.0  | 44.0   | 94.0    | 60.0   | 76.0    | 46.0   | 50.0    |
> | GPT-5.1            | 100.0| 94.0  | 80.0 | 50.0  | 86.0 | 92.0  | 60.0   | 100.0   | 72.0   | 92.0    | 56.0   | 44.0    |
> | GPT-4.1            | 98.0 | 82.0  | 44.0 | 44.0  | 84.0 | 88.0  | 50.0   | 12.0    | 46.0   | 34.0    | 34.0   | 32.0    |
>
> As shown in the table, the advanced models demonstrated significant performance gains, particularly in QT4 (Temporal Scheduling). While this is encouraging, these performance gains come with a critical trade-off in latency. We measured the average episode completion time (in seconds) for high-performing reasoning models across each QT. All values are reported in seconds.
>
> | Models           | Avg. | QT1-F | QT1-IF | QT2-F | QT2-IF | QT3-F | QT3-IF | QT4-1-F | QT4-1-IF | QT4-2-F | QT4-2-IF | QT4-3-F | QT4-3-IF |
> |-----------------|------|-------|--------|-------|--------|-------|--------|---------|----------|---------|----------|---------|----------|
> | Gemini2.5-pro  | 44.6 | 24.1 | 22.4  | 57.5 | 48.8  | 66.1 | 27.8  | 74.0   | 12.5    | 57.7   | 37.0    | 53.7   | 53.1    |
> | GPT-5.1         | 82.1 | 35.7 | 38.4   | 109.4| 99.6   | 78.6  | 54.3   | 121.1   | 13.5    | 135.1  | 76.0    | 112.7  | 111.0   |
> | GPT-4.1         | 19.9 | 8.3  | 7.8    | 23.6 | 20.2   | 22.9  | 9.4    | 26.6    | 12.3    | 28.7   | 23.7    | 29.7   | 25.9    |
>
> According to the results, the large reasoning models have unacceptably large latency for practical deployment; GPT-5.1 takes up to 135.1 seconds and Gemini2.5-pro takes up to 74 seconds on average. In a smart home context, users typically expect immediate responses in less than a second. Therefore, the extended latency limits practical utility even when accuracy improves.
>
> At the other end of the spectrum, small models under 7B parameters, which are more practical choices in deployment, achieved minimal performance. This suggests that the complex reasoning and tool-use coordination required by SimuHome present significant challenges for resource-constrained models.
>
> These results highlight a critical trade-off between reasoning accuracy and response latency, and reveal a significant capability gap across model scales. We hope that SimuHome serves as a valuable testbed to help diagnose this trade-off in practical smart home deployment scenarios.
>
> **We have incorporated these experiments and analysis in Section 5.1 and Section 5.2.3 of the revision.**

---

> ### Author Response · Authors · 2025-11-26
>
> > **W2.1. Table 1 is confusing, especially regarding the different superscripts J and S. In QT1, both F and IF are marked with J (for LLM-judge-based evaluation), but for the following queries, the format seems to be inconsistent.**
>
> **A2.1.** We apologize for the unclear notation in Table 1. To clarify, QT1 uses LLM-judge evaluation for both feasible and infeasible episodes because success depends on whether the agent provides accurate information in natural language, rather than changing physical device states. In contrast, QT2-F, QT3-F, and QT4-F require actual state changes, so we use simulator-based evaluation. **We have incorporated this clarification into Section 4.3 of the revision**, explicitly specifying the evaluation protocols for each query type and explaining the rationale for using the LLM-judge in QT1.
>
> &nbsp;
> > **W2.2. Also, in Table 1 for the QT2-F Task, the success rate of GPT-4.1 is 44%, which is far behind GPT-4.1-mini and Gemini-2.5-Flash. I'm curious about the reason for this phenomenon.**
>
> **A2.2.** Regarding GPT-4.1's lower performance on QT2-F (44%), the root cause lies in the transition_time parameter for dimmable lights, which specifies the duration for brightness changes. GPT-4.1-mini and Gemini2.5-Flash set this parameter to 0 seconds for immediate brightness changes, while GPT-4.1 set it to 2-3 seconds for gradual transitions. We opt to verify the home states immediately after task completion because the queries do not request gradual transitions and it is better to avoid unexpected environment changes that may interfere with the lights during the transition. As a result, GPT-4.1 had not yet reached the target brightness at evaluation time.
> When we allow a 3-second delay, GPT-4.1's success rate increases to 62% from 44%.
>
> **We have incorporated this analysis in Appendix I of the revision.**

---

> > ### Comment · Reviewer_hnkK · 2025-11-26
> >
> > Thanks for providing detailed additional experiments to address my concerns. The added experiments show some interesting results that worth further exploration.
> >
> > I've raised my score accordingly.  Good Luck.

---

> > > ### Author Response · Authors · 2025-12-03
> > >
> > > We are deeply grateful for your thorough re-evaluation and the substantial score increase from 4 to 8.

---

### Official Review · Reviewer_wJtQ · 2025-10-30

**Soundness:** 3
**Presentation:** 4
**Contribution:** 4
**Rating:** 8
**Confidence:** 3

**Summary:**

The paper introduces SimuHome, a high-fidelity simulator and benchmark designed to evaluate LLM agents in smart home environments. The simulator is Matter-protocol compliant, supporting real-time device interactions, temporal dependencies, and environmental feedback (temperature, humidity, etc.).

The benchmark includes 600 manually validated episodes across 12 query types, including environment perception, implicit intent, explicit intent, and three complex types of temporal scheduling (future, dependency, and concurrent). A key feature is that each type includes both "feasible" and "infeasible" variants to test an agent's robustness and logical reasoning.

The authors evaluated 11 prominent LLM agents under a unified ReAct framework. The results clearly show that while models can handle simple, explicit commands, they severely struggle with inferring latent intent, verifying states, and especially handling complex temporal scheduling (QT4). Even the top-performing model, GPT-4.1, achieved an overall accuracy of only 54%.

**Strengths:**

1.	High Fidelity in simulator design and Matter integration.
2.	The benchmark design is excellent. The 12 query types cover a wide range of scenarios, from simple perception to complex scheduling. The introduction of "feasible" vs. "infeasible" variants is crucial for testing an agent's "refusal" capabilities and logical consistency.
3.	The focus on temporal reasoning (QT4) is the paper's biggest highlight. It correctly identifies this as the "Achilles' heel" of current SOTA models, including GPT-4.1. The error analysis, which distinguishes between "Contradiction Mishandling (CM)" and "Contradiction Blindness (CB)," is highly insightful for understanding the root cause of these failures.
4.	The analysis in Section 6.2 is brilliant. The authors found that agents perform well on QT3 (Explicit Control) because they can "passively learn" and recover from the tool's immediate error feedback. In contrast, they fail on QT4 (Temporal Scheduling) because the schedule_workflow tool only returns a "scheduling successful" acknowledgment, and the feedback is deferred (the failure only becomes apparent at execution time). This points to a clear direction for future research (e.g., the need for "pre-validation" tools).

**Weaknesses:**

1.	The paper mentions simulating 4 environmental variables (temperature, illuminance, humidity, air quality) and that device operations have a cumulative impact. However, the current design seems focused on a one-way "device→environment" effect. The paper does not detail whether more complex interactions are modeled, such as "environment→environment" (e.g., does an open window affect the AC's cooling efficiency?) or "device→device" (e.g., does the simulator model power load conflicts, such as 'tripping the breaker' as hinted at in the QT4-3 example?).
2.	The benchmark consists of 600 fixed, human-validated episodes. While high-quality, is this sufficient to prevent SOTA models (especially closed-source ones) from "memorizing" the 12 query types after a few iterations? The authors mention generating layouts to "prevent agent overfitting", but this seems to refer to the layout, not the task itself. A discussion on the potential for dynamic generation or scalability of the benchmark would be beneficial.
3.	The evaluation is standardized on the ReAct framework, which is good for a fair comparison. However, ReAct is a relatively simple "think-act" loop. The paper attributes the failures in QT4 to the models, but to what extent is this also a failure of the ReAct framework itself, which is not inherently designed for long-term planning? Would agents using more complex planning algorithms (e.g., Tree-of-Thoughts or a dedicated planning loop) perform differently?

**Questions:**

1.	Can you elaborate on the complexity of the environmental interactions? Does the SimuHome simulator model "environment→environment" or "device→device"   interactions?
2.	The evaluation standardizes on the ReAct framework. How much of the significant failure on QT4 (temporal) tasks do you attribute to the models' core reasoning limitations versus the inherent limitations of the ReAct framework for complex, long-term planning?
3.	Your analysis in Section 6.2 regarding "deferred feedback" is very insightful. Based on this, what do you believe is the most promising path forward: developing better "pre-validation" tools to provide immediate feedback, or creating more advanced agent architectures that can reason about deferred outcomes?

---

> ### Author Response · Authors · 2025-11-26
>
> We deeply appreciate your high assessment of our contribution and your clear recognition of our core research intent. We are particularly encouraged by your focus on the value of our “feasible vs. infeasible” design and the critical insight into how deferred feedback in temporal scheduling limits agent learning.
> We have addressed your specific questions below.
> ___
> > **W1. The paper mentions simulating 4 environmental variables (temperature, illuminance, humidity, air quality) and that device operations have a cumulative impact. However, the current design seems focused on a one-way "device$\rightarrow$environment" effect. The paper does not detail whether more complex interactions are modeled, such as "environment$\rightarrow$environment" (e.g., does an open window affect the AC's cooling efficiency?) or "device$\rightarrow$device" (e.g., does the simulator model power load conflicts, such as 'tripping the breaker' as hinted at in the QT4-3 example?).**
>
> > **Q1. Can you elaborate on the complexity of the environmental interactions? Does the SimuHome simulator model "environment$\rightarrow$environment" or "device$\rightarrow$device" interactions?**
>
> **A1.** Thank you for this thought-provoking question.
> We fully agree that modeling complex interactions between environments (Environment $\rightarrow$ Environment) or between devices (Device $\rightarrow$ Device) would significantly enhance the realism of the simulation.
> In fact, our Aggregator architecture is designed to be highly extensible, and the suggested interactions can be quite easily implemented in our framework by introducing additional devices that mediate environmental variables (environment→environment) and additional environmental variables that mediate devices (device→device), along with updated equations for environmental variables such as temperature or illuminance.
>
> Below are potential implementation approaches for the specific examples you suggested.
>
> **Environment → Environment interaction (e.g., open window):** A window can be modeled as a standard device with Open/Close/OutsideTemperature attributes that directly affects indoor temperature (e.g., to be closer to the outside temperature). By adding an external heat influx coefficient to the current Aggregator equation, we can dynamically reduce the cooling efficiency of the air conditioner when the window is in the "Open" state.
>
> **Device → Device interaction (e.g., power load):** Similarly, "total power load" can be defined and tracked as an environmental variable, analogous to temperature or illuminance, whose value is adjusted by the power consumption of devices in the home. This variable can then mediate interactions between devices. For example, if the total power load exceeds a safety threshold, the simulator can forcibly shut down all devices, thereby simulating a breaker trip scenario.
>
> **We have incorporated this discussion in Appendix K of the revision.**
> We plan to incorporate these ideas in future simulator updates.
> Thank you for suggesting the great idea.

---

> ### Author Response · Authors · 2025-11-26
>
> > **W2. The benchmark consists of 600 fixed, human-validated episodes. While high-quality, is this sufficient to prevent SOTA models (especially closed-source ones) from "memorizing" the 12 query types after a few iterations? The authors mention generating layouts to "prevent agent overfitting", but this seems to refer to the layout, not the task itself. A discussion on the potential for dynamic generation or scalability of the benchmark would be beneficial.**
>
> **A2.** To empirically investigate the potential memorization of the 12 query types, we have conducted an additional SFT experiment.
> First, we constructed a high-quality training dataset by compiling 204 gold trajectories (17 per type) that GPT-5.1 successfully solved on newly generated episodes distinct from the original benchmark.
>
> For the experiment, we evaluated small-scale models (<7B) such as Llama3.2-1B/3B-it and Gemma3-4B-it (Table 1) due to time constraints. We selected Gemma3-4B-it because it was the only model that recorded non-zero performance on the main tasks. A non-zero baseline is essential to meaningfully measure the impact of SFT.
>
> Table 1. Performance of tested models (%)
> | Models              | QT1-F | QT1-IF | QT2-F | QT2-IF | QT3-F | QT3-IF | QT4-1-F | QT4-1-IF | QT4-2-F | QT4-2-IF | QT4-3-F | QT4-3-IF |
> |--------------------|-------|--------|-------|--------|-------|--------|---------|----------|---------|----------|---------|----------|
> | Llama3.2-1b-inst  | 0.0  | 0.0   | 0.0  | 0.0   | 0.0  | 0.0   | 0.0    | 0.0     | 0.0    | 0.0     | 0.0    | 0.0     |
> | Llama3.2-3b-inst  | 10.0 | 12.0  | 0.0  | 2.0   | 4.0  | 0.0   | 2.0    | 0.0     | 2.0    | 0.0     | 0.0    | 0.0     |
> | Gemma3-4b-it      | 44.0 | 32.0  | 12.0 | 10.0  | 28.0 | 8.0   | 0.0    | 0.0     | 2.0    | 0.0     | 0.0    | 4.0     |
>
> Table 2. Performance of fine-tuned Gemma3-4b-it (%)
> | Models              | QT2-F | QT3-F | QT4-1-F | QT4-2-F | QT4-3-F |
> |--------------------|-------|-------|---------|---------|---------|
> | Gemma3-4b-it      | 12.0 | 28.0 | 0.0    | 2.0    | 0.0    |
> | Gemma3-4b-it SFT  | 22.0 | 24.0 | 4.0    | 4.0    | 0.0    |
>
> The results show mixed outcomes: while QT2-F improved from 12% to 22%, QT3-F decreased from 28% to 24%, and temporal reasoning tasks (QT4) remained near zero.
> Despite explicit training on gold trajectories, the model failed to generalize to dynamic environmental variations. This demonstrates that the tasks in our benchmark cannot be accomplished by simply memorizing successful trajectories, and static datasets alone have clear limitations in handling dynamic environmental changes.
>
> Although we have reported the fine-tuning results only on small models due to time constraints, we plan to extend experiments to models of various sizes and families in the camera-ready version.
> **The above results have been incorporated in Appendix H of the revision.**

---

> ### Author Response · Authors · 2025-11-26
>
> > **W3. The evaluation is standardized on the ReAct framework, which is good for a fair comparison. However, ReAct is a relatively simple "think-act" loop. The paper attributes the failures in QT4 to the models, but to what extent is this also a failure of the ReAct framework itself, which is not inherently designed for long-term planning? Would agents using more complex planning algorithms (e.g., Tree-of-Thoughts or a dedicated planning loop) perform differently?**
>
> > **Q2. The evaluation standardizes on the ReAct framework. How much of the significant failure on QT4 (temporal) tasks do you attribute to the models' core reasoning limitations versus the inherent limitations of the ReAct framework for complex, long-term planning?**
>
> **A3.**
> We fully agree with your point that ReAct may have structural disadvantages for longer horizon tasks. In our initial experiments, we adopted ReAct as a standard framework in agent research to establish a universal baseline for benchmark performance.
> To disentangle the limitations of the framework from the model’s core capabilities, we conducted two additional ablation studies: 1) Adopting a complex planning algorithm and 2) Enabling Runtime Self-Review.
>
> **[Experiment 1: Adopting a Complex Planning Algorithm]**
> Based on your suggestion, we adopted HiAgent (Hu et al., 2025), which employs a hierarchical planning algorithm that actively generates sub-goals and summarizes completed trajectories. We then conducted comparative experiments on QT4 using this framework.
> | Models             | QT4-1-F | QT4-2-F | QT4-3-F |
> |-------------------|---------|---------|---------|
> | GPT-4.1 + ReAct   | 50.0%   | 46.0%   | 34.0%   |
> | GPT-4.1 + HiAgent | 40.0%   | 52.0%   | 40.0%   |
>
> As shown, HiAgent outperforms ReAct on QT4-2 and QT4-3 (52% vs 46%, 40% vs 34%), but underperforms on QT4-1 (40% vs 50%). This indicates that increasing planning complexity alone does not fully resolve temporal reasoning failures and can even introduce overhead in simpler constraints.
>
> **[Experiment 2: Enabling Runtime Periodic Self-Review]**
> **Settings:** To investigate further, we also tested a realistic setting where agents can adjust plans based on runtime state through periodic self-review, effectively mitigating the limitation of our initial setup where agents are unable to modify their schedules after the initial planning phase.
>
> Specifically, after scheduling workflows, the agent receives callback triggers every 5 minutes (from the simulator) up to and including the execution time. During each check, the agent reviews scheduled workflows using `get_workflow_list` and may inspect the home state using other tools, and executes corrections using `cancel_workflow` if needed. The callback also notifies the agent to check the home state immediately after the scheduled task is executed, but it does not provide an explicit notice of success or failure. Therefore, the agent is required to independently evaluate the outcome to determine the appropriate subsequent actions. This setup strictly tests the agent's ability to autonomously detect and correct planning errors.
>
> **Results:**
> | Query Type | Failed Cases | Recovery Success | Recovery Rate | Avg. Steps |
> |------------|--------------|------------------|----------------|------------|
> | QT4-1      | 25           | 2                | 8.0 %          | 64.44      |
> | QT4-2      | 27           | 5                | 18.5 %         | 26.80      |
> | QT4-3      | 33           | 0                | 0.0 %         | 29.67      |
>
> Self-review achieves a negligible recovery rate of only 0–18.5%, despite incurring high computational costs (26–64 steps). Crucially, even when the agent was explicitly prompted to inspect the home state immediately after execution, it failed to recognize the task failure in most cases. This demonstrates that enabling the agent to periodically check the home state both before and immediately after execution does not yield significant performance improvements.
>
> The continued failure on QT4 tasks, even after adopting advanced planning techniques suitable for long horizons (Experiment 1) and enabling runtime periodic self-review (Experiment 2), demonstrates that the primary bottleneck is not the ReAct framework itself.
> Instead, the failures are likely attributed to the inherently limited capabilities of current models to ground complex home states and perform precise temporal reasoning.
>
> **We have incorporated these experiments in Section 5.2.4 of the revision.**
>
> &nbsp;
> ___
> [1] Hu et al. (2025). HiAgent: Hierarchical Working Memory Management for Solving Long-Horizon Agent Tasks with Large Language Model. ACL 2025.

---

> ### Author Response · Authors · 2025-11-26
>
> > **Q3. Your analysis in Section 6.2 regarding "deferred feedback" is very insightful. Based on this, what do you believe is the most promising path forward: developing better "pre-validation" tools to provide immediate feedback, or creating more advanced agent architectures that can reason about deferred outcomes?**
>
> **A4.** We highly appreciate your question on deferred feedback.
> We believe both directions have merit, but the most promising path forward may be to integrate SimuHome directly into the agent architecture as a runtime “world model” for pre-validation, going beyond simple API checkers.
> Pre-validation is a non-trivial task because feasibility in smart home environments is not determined by fixed rules but varies depending on dynamic state changes and interactions between devices. For instance, a scheduled workflow that was considered valid at the registration time could become invalid and lead to execution failure if other events occur between scheduling and execution time and conditions change. Therefore, we believe that running simulations is crucial and that agent reasoning alone may not be sufficient to account for the complexity of dynamic environments.
>
> Specifically, in a real-world deployment scenario, upon receiving a scheduling request, the agent would first execute the plan within SimuHome. By leveraging SimuHome’s time acceleration capability, the agent can immediately observe future outcomes and detect potential conflicts. If issues arise, the agent revises the plan within the simulation before committing to the real-world action. Furthermore, simulations can be conducted periodically to enable the agent to detect potential execution errors in advance. This approach combines the benefits of both pre-validation and architectural advances by embedding simulation-based reasoning directly into the agent’s decision-making process.
>
> **We have incorporated this discussion into Appendix J of the revision.**

---

> > ### Comment · Reviewer_wJtQ · 2025-11-27
> >
> > Thank you for the detailed response. I will maintain my positive score.

---

> > > ### Author Response · Authors · 2025-12-03
> > >
> > > Thank you for your strong support and positive evaluation.

---

### Official Review · Reviewer_zEmB · 2025-10-30

**Soundness:** 3
**Presentation:** 3
**Contribution:** 3
**Rating:** 6
**Confidence:** 4

**Summary:**

The author introduce SimuHome, a time-accelerated home environment that simulates smart devices, supports API calls, and reflects changes in environmental variables. The benchmark contains test realistic query in smart home scenario like “When the dishwasher finishes, please turn off the kitchen lights”.

**Strengths:**

* The simulator is a useful artifact for future work.

* The analysis in the work provides an insightful look into model's behavior.

* Reproducible simulation environment

**Weaknesses:**

* The baseline in the model is using ReAct which is disadvantaged for longer horizon tasks like QT4. Have the authors think about including a memory-augmented baseline?

* Could the bad performance on infeasible test cases be solvable if prompted properly?

**Questions:**

* Line 429, the lack of tool feedback. Should it be a feature that needs to be added to the simulator or something tested system needs to deal with?

* How is home layout presented to LLM?

* For queries like "When the dishwasher finishes, please turn off the kitchen lights", how much of it could be resolved by having a system that set a callback and prompt LLM when the time comes?

---

> ### Author Response · Authors · 2025-11-26
>
> We deeply appreciate your recognition that SimuHome provides a reproducible testbed for the community and that our analysis offers valuable insights into model behavior. We have addressed your specific questions below.
> ___
> > **W1. The baseline in the model is using ReAct which is disadvantaged for longer horizon tasks like QT4. Have the authors think about including a memory-augmented baseline?**
>
> **A1.** We initially adopted ReAct because it is the most widely used, standard framework in agent research, and it is important to establish a universal baseline for benchmark performance. However, we acknowledge that memory-augmented methods may be better suited for long-horizon tasks. Based on your suggestion, we have adopted HiAgent (Hu et al., 2025) as a memory-augmented baseline. HiAgent employs a 'Hierarchical Working Memory' system that compresses completed sub-goals into summaries to reduce context load. We then conducted comparative experiments on QT4 using this framework.
>
> | Models             | QT4-1-F | QT4-2-F | QT4-3-F |
> |-------------------|---------|---------|---------|
> | gpt-4.1 + ReAct   | 50.0%   | 46.0%   | 34.0%   |
> | gpt-4.1 + HiAgent | 40.0%   | 52.0%   | 40.0%   |
>
> HiAgent outperforms ReAct on QT4-2 and QT4-3 (52% vs 46%, 40% vs 34%), but underperforms on QT4-1 (40% vs 50%). These results indicate that while ReAct's framework has limited support for complex temporal coordination, not all temporal reasoning failures stem from the framework. Even with sophisticated planning, precise timestamp calculation remains a fundamental challenge.
>
> &nbsp;
> ___
> [1] Hu et al. (2025). HiAgent: Hierarchical Working Memory Management for Solving Long-Horizon Agent Tasks with Large Language Model. ACL 2025.

---

> ### Author Response · Authors · 2025-11-26
>
> > **W2. Could the bad performance on infeasible test cases be solvable if prompted properly?**
>
> **A2.** Thank you for raising this point.
> We indeed systematically optimized the agent’s system prompt through iterative testing, and the current version in the manuscript represents the best-performing configuration across all query types. Each instruction under the [VERIFICATION & ACCURACY] section in the latest prompt (Appendix O.1) was added to address specific failure patterns observed during preliminary experiments. For example, when agents blindly trusted user statements about device existence, we added the instruction to use get_room_devices for verification. When agents confused temporal information, we added the instruction to use get_current_time for confirmation. The final system prompt includes explicit verification requirements for infeasible scenarios.
> Despite these explicit, targeted instructions developed through iterative refinement, agents still struggled with infeasible scenarios.
>
> &nbsp;
> > **Q1. Line 429, the lack of tool feedback. Should it be a feature that needs to be added to the simulator or something tested system needs to deal with?**
>
> **A3.** We view handling the lack of immediate tool feedback as a core capability that the agent should possess.
> In smart home environments, feasibility is not determined by simple rules but must be inferred based on dynamic state changes and interactions between devices.
> To make things more difficult, a scheduled workflow that was considered valid at the registration time could become invalid and lead to execution failure if other events occur between scheduling and execution time and conditions change. Therefore, the SimuHome simulator itself has no bullet-proof solution, and we do not recommend agents rely solely on the simulator's imperfect validation.
> Instead, the agent should proactively verify preconditions and anticipate potential failures before scheduling. To support this, we provide tools such as `get_workflow_list` and `cancel_workflow` that allow the agent to audit and refine its own plan.
>
> &nbsp;
> > **Q2. How is home layout presented to LLM?**
>
> **A4.** The agent dynamically perceives the home layout and devices by using tools such as `get_rooms` and `get_room_devices`. This design reflects realistic deployment scenarios where agents cannot assume prior knowledge of the home configuration, which could be changed over time, and must discover the environment through tool use. In our model prompt, we explicitly instruct the model to use `get_rooms`, `get_room_devices`, and `get_room_states` for each user request.
> The full list of available tools is provided in Appendix B.

---

> ### Author Response · Authors · 2025-11-26
>
> > **Q3. For queries like "When the dishwasher finishes, please turn off the kitchen lights", how much of it could be resolved by having a system that set a callback and prompt LLM when the time comes?**
>
> **A5.** Based on your suggestion, we implemented a callback-based feedback loop to evaluate whether agents can resolve scheduling failures when notified at the scheduled time.
>
> **[Experiment 1: Resolving failure via Callback]**
> **Settings:** We implemented a multi-turn feedback loop to test post-execution re-evaluation when a failure is explicitly reported. At the scheduled execution time, the SimuHome simulator checks whether the target device state was achieved. If not, a user simulator (GPT-5-mini) provides natural language feedback to the agent (e.g., "The device you scheduled is not in the expected state"). The agent then re-attempts the task with this feedback.
> It is important to note that this setting represents a highly idealized scenario that is less realistic for real-world deployment. In real scenarios, it is difficult to expect an oracle to immediately notify the agent of an execution failure at the scheduled time. Instead, the agent is expected to inspect the success or failure of the scheduled task without the involvement of an oracle or the user.
> **Results:**
> | Query Type | Failed Cases | Recovery Success | Recovery Rate | Avg. Steps |
> |------------|--------------|------------------|----------------|------------|
> | QT4-1      | 25           | 15               | 60.0          | 6.7        |
> | QT4-2      | 27           | 15               | 55.6%          | 4.7        |
> | QT4-3      | 33           | 22               | 66.7%          | 4.4        |
>
> With explicit post-execution failure notice, the agent recovered in 55-67% of failed cases with efficient step counts (4.4-6.7 steps). However, these results should be interpreted with caution as they represent an upper bound on performance in this highly idealized scenario. The high recovery rate is not practically achievable in real deployments, as failure notices require oracle supervision. Moreover, for users, the initial failure is still perceived as a failure, even if the model can correct it after the user's complaint.
>
> **[Experiment 2: Runtime Plan Adjustment through Periodic Self-Review]**
> **Settings:** To assess a more realistic setting, we tested whether agents can adjust plans based on runtime state through periodic self-review. After scheduling workflows, the agent receives callback triggers every 5 minutes (from the simulator) up to and including the execution time. During each check, the agent reviews scheduled workflows using `get_workflow_list` and may inspect the home state using other tools, and can cancel and reschedule incorrect workflows using `cancel_workflow`. Importantly, the callback notifies the agent to check the home state immediately after the scheduled task runs, but it does not notify whether the intended goal state has been met or not. Therefore, the agent is required to independently determine the success or failure of the task to proceed with subsequent actions. This tests the agent's ability to autonomously detect and correct planning errors before and right after execution.
> **Results:**
> | Query Type | Failed Cases | Recovery Success | Recovery Rate | Avg. Steps |
> |------------|--------------|------------------|----------------|------------|
> | QT4-1      | 25           | 2                | 8.0 %          | 64.44      |
> | QT4-2      | 27           | 5                | 18.5 %         | 26.80      |
> | QT4-3      | 33           | 0                | 0.0 %         | 29.67      |
>
> Self-review achieves only 0-18% recovery, and this relatively low recovery rate comes at high computational costs (26-64 steps). The result shows that enabling the agent to periodically check the home state before the scheduled time and immediately after the scheduled execution does not lead to significant improvement in task performance. Hence, the low performance of models on QT4 is likely attributed to the inherently limited capabilities of these models to solve complex tasks based on the home state, rather than to the number of assessment opportunities provided.
>
> **We have incorporated these experiments in Section 5.2.4 and Appendix G of the revision.**

---

### Official Review · Reviewer_qKgg · 2025-11-01

**Soundness:** 3
**Presentation:** 3
**Contribution:** 3
**Rating:** 6
**Confidence:** 3

**Summary:**

This paper introduces SimuHome, a Matter protocol-based smart home simulator and benchmark system for evaluating Large Language Model agents in realistic home environments. The system simulates 17 device types and 4 environmental variables, providing 600 test episodes covering 12 query types including environment perception, intent inference, device control, and temporal scheduling. Evaluation of 11 mainstream LLMs reveals that the best-performing model, GPT-4.1, achieves only 54% overall accuracy, with temporal scheduling tasks being the most challenging.

**Strengths:**

1. Smart home control addresses real-world challenges in production systems, including latent intent inference, temporal dependencies, and device constraints, making the research practically valuable.
2. The benchmark covers 12 diverse query types with 600 carefully validated episodes featuring both feasible and infeasible scenarios.
3. The paper evaluates 11 diverse models with detailed error analysis using a well-defined taxonomy, examines tool-call patterns and error recovery mechanisms, providing valuable insights into current limitations and improvement directions.

**Weaknesses:**

1. Limited model diversity in evaluation, particularly lacking small-parameter models (e.g., <7B parameters) that would be practically relevant for on-device smart home scenarios where computational resources are constrained and real-time response is critical.
2. The benchmark lacks support for multi-turn interactive dialogues and clarification exchanges. Real-world smart home interactions often involve users providing additional context or correcting misunderstandings across multiple turns, which is not captured in the current single-turn episode design.
3. The temporal scheduling mechanism is overly restrictive, requiring agents to perform static planning without the ability to schedule periodic tasks, dynamically re-evaluate conditions, or adjust plans based on runtime state changes. This limits the realism of temporal reasoning evaluation compared to real smart home systems that support event-driven and adaptive scheduling.

**Questions:**

See weaknesses.

---

> ### Author Response · Authors · 2025-11-26
>
> We sincerely appreciate your thoughtful feedback, particularly your recognition of SimuHome's practical value in addressing real-world smart home challenges and the rigor of our evaluation methodology. We have addressed your specific questions below.
> ___
> > **W1. Limited model diversity in evaluation, particularly lacking small-parameter models (e.g., <7B parameters) that would be practically relevant for on-device smart home scenarios where computational resources are constrained and real-time response is critical.**
>
> **A1**. We fully agree that evaluating smaller models is essential given the computational constraints of real-world deployment environments. Based on your feedback, we have additionally evaluated Llama3.2-1B-it, Llama3.2-3B-it, and Gemma3-4B-it. For comprehensiveness, we also included advanced reasoning models (Gemini2.5-Pro and GPT-5.1) to provide a more holistic comparison across model scales. The results are shown in the table below. Success rates in the table are presented as percentages.
>
> | Models              | QT1-F | QT1-IF | QT2-F | QT2-IF | QT3-F | QT3-IF | QT4-1-F | QT4-1-IF | QT4-2-F | QT4-2-IF | QT4-3-F | QT4-3-IF |
> |--------------------|-------|--------|-------|--------|-------|--------|---------|----------|---------|----------|---------|----------|
> | llama3.2-1b-it  | 0.0  | 0.0   | 0.0  | 0.0   | 0.0  | 0.0   | 0.0    | 0.0     | 0.0    | 0.0     | 0.0    | 0.0     |
> | llama3.2-3b-it  | 10.0 | 12.0  | 0.0  | 2.0   | 4.0  | 0.0   | 2.0    | 0.0     | 2.0    | 0.0     | 0.0    | 0.0     |
> | gemma3-4b-it      | 44.0 | 32.0  | 12.0 | 10.0  | 28.0 | 8.0   | 0.0    | 0.0     | 2.0    | 0.0     | 0.0    | 4.0     |
> | gemini2.5-pro     | 96.0 | 78.0  | 60.0 | 56.0  | 76.0 | 72.0  | 44.0   | 94.0    | 60.0   | 76.0    | 46.0   | 50.0    |
> | gpt-5.1            | 100.0| 94.0  | 80.0 | 50.0  | 86.0 | 92.0  | 60.0   | 100.0   | 72.0   | 92.0    | 56.0   | 44.0    |
> | gpt-4.1            | 98.0 | 82.0  | 44.0 | 44.0  | 84.0 | 88.0  | 50.0   | 12.0    | 46.0   | 34.0    | 34.0   | 32.0    |
>
> The small-scale models (Llama3.2-1B-it, Llama3.2-3B-it, and Gemma3-4B-it) exhibit low accuracy and performance degradation compared to larger models in the paper. This is expected because even the existing 12B model in the manuscript (Gemma3-12B-it) struggled with complex reasoning. While Gemma3-4B-it notably achieved relatively meaningful performance, it still revealed limitations in complex temporal reasoning (QT4). These results suggest that the higher-order reasoning capabilities required by SimuHome remain limited in current small-scale models.
>
> On the other hand, the advanced reasoning models (Gemini-2.5-Pro and GPT-5.1) demonstrated significant performance gains, particularly in QT4. While this is encouraging, these gains come with a critical trade-off in latency. We measured the average episode completion time (in seconds) for high-performing models (including the best standard model, GPT-4.1) across each QT. All values are reported in seconds.
>
> | Models           | Avg. | QT1-F | QT1-IF | QT2-F | QT2-IF | QT3-F | QT3-IF | QT4-1-F | QT4-1-IF | QT4-2-F | QT4-2-IF | QT4-3-F | QT4-3-IF |
> |-----------------|------|-------|--------|-------|--------|-------|--------|---------|----------|---------|----------|---------|----------|
> | gemini2.5-pro  | 44.6 | 24.1 | 22.4  | 57.5 | 48.8  | 66.1 | 27.8  | 74.0   | 12.5    | 57.7   | 37.0    | 53.7   | 53.1    |
> | gpt-5.1         | 82.1 | 35.7 | 38.4   | 109.4| 99.6   | 78.6  | 54.3   | 121.1   | 13.5    | 135.1  | 76.0    | 112.7  | 111.0   |
> | gpt-4.1         | 19.9 | 8.3  | 7.8    | 23.6 | 20.2   | 22.9  | 9.4    | 26.6    | 12.3    | 28.7   | 23.7    | 29.7   | 25.9    |
>
> As the reviewer also alluded to in the question, large reasoning models have unacceptably large latency for practical deployment; GPT-5.1 takes up to 135.1 seconds and Gemini-2.5-pro takes up to 74 seconds on average, while GPT-4.1 takes up to 29.7 seconds. In a smart home context, users typically expect immediate responses in less than a second. Therefore, even GPT-4.1 (the best-performing standard model) cannot meet real-time requirements, and the reasoning models’ extended latency further limits practical utility despite their accuracy improvements.
>
> At the other end of the spectrum, small-scale models under 7B parameters, which are more practical choices in deployment, achieved minimal performance. This suggests that SimuHome’s complex reasoning and tool-use coordination present significant challenges for resource-constrained models.
> These results highlight a critical trade-off between reasoning accuracy and response latency, and reveal a significant capability gap across model scales. We hope that SimuHome serves as a valuable testbed to help diagnose this trade-off in practical smart home deployment scenarios.
>
> **We have incorporated these results in Section 5.1 and Section 5.2.3 of our revision.**

---

> ### Author Response · Authors · 2025-11-26
>
> >**W2. The benchmark lacks support for multi-turn interactive dialogues and clarification exchanges. Real-world smart home interactions often involve users providing additional context or correcting misunderstandings across multiple turns, which is not captured in the current single-turn episode design.**
>
> **A2.** We fully recognize the importance of providing additional context or correcting misunderstandings through multi-turn interactions in real-world smart home environments. Based on your feedback, we implemented two new multi-turn experimental settings using the QT2-F dataset, because it concerns queries with latent user intents (please note that additional experiments on QT4 will be discussed in our response to W3). In these experiments, we used the GPT-4.1 model.
>
> **[Experiment 1: Multi-turn Providing Context]**
> **Settings:** To address your point about users providing clarifications and additional context, we built a user simulator based on GPT-4.1-mini designed to provide goal-aligned information when requested. We enabled the agent to ask for clarification from the user through an ask_user() tool and configured the prompt to encourage its use when information is uncertain.
> **Results:** The success rate increased slightly from 44% to 50%. However, the model did not sufficiently utilize the `ask_user` tool, despite our explicit prompt to use this action when clarification is needed. Only 10 out of 28 failed cases called the `ask_user` tool. We attribute this to the model's inability to recognize situational ambiguity on its own.
>
>
> **[Experiment 2: Multi-turn Correcting Misunderstandings]**
> **Settings:** To address your point about correcting misunderstandings across multiple turns, we implemented a correction loop where, if the agent fails to complete a task, the user simulator provides explicit feedback such as "Incorrect. Please review and try again". Please note that users are highly likely to perceive such cases as failures anyway.
> **Results:** The success rate improved from 44% to 54%. However, despite receiving explicit feedback, the failure rate remained high. This suggests that the model's self-correction capability, the ability to diagnose and fix its own reasoning errors, is still limited.
>
> Overall, our multi-turn experiments confirmed that conversational interaction improves performance from 44% to 50-54%. Unfortunately, the performance plateau at this level indicates that multi-turn interaction remains insufficient to fully address the underlying challenges. Our results suggest that the low performance of models is not merely due to insufficient interaction with the user, but to their lack of fundamental capabilities.
>
> Hence, although our current benchmark focuses on single-turn episodes, we hope that this benchmark serves as a valuable resource for enhancing models' core cognitive abilities for ambiguity detection in Feasible scenarios and contradiction identification in Infeasible scenarios. We also believe that our work can serve as a solid basis for an extension to multi-turn interactive dialogues in future work.
>
> **We have incorporated the above multi-turn experiment results in Appendix F of the revision.**

---

> ### Author Response · Authors · 2025-11-26
>
> > **W3. The temporal scheduling mechanism is overly restrictive, requiring agents to perform static planning without the ability to schedule periodic tasks, dynamically re-evaluate conditions, or adjust plans based on runtime state changes. This limits the realism of temporal reasoning evaluation compared to real smart home systems that support event-driven and adaptive scheduling.**
>
> **A3.** Following your suggestion about dynamic re-evaluation and runtime plan adjustment, we have conducted two additional experiments to examine whether agents can recover from scheduling failures through these mechanisms. Both experiments focused on episodes where GPT-4.1 initially failed on QT4-1, QT4-2, and QT4-3-F.
>
> **[Experiment 1: Runtime Plan Adjustment through Periodic Self-Review]**
> **Settings:** We tested whether agents can adjust plans based on runtime state through periodic self-review. After scheduling workflows, the agent performs checks every 5 minutes (triggered by the simulator) up to and including the execution time. During each check, the agent reviews scheduled workflows using `get_workflow_list` and may inspect the home state using other tools, and can cancel and reschedule incorrect workflows using `cancel_workflow` tool. This tests the agent's ability to detect and correct planning errors before and right after execution.
> **Results:**
> | Query Type | Failed Cases | Recovery Success | Recovery Rate | Avg. Steps |
> |------------|--------------|------------------|----------------|------------|
> | QT4-1      | 25           | 2                | 8.0 %          | 64.44      |
> | QT4-2      | 27           | 5                | 18.5 %         | 26.80      |
> | QT4-3      | 33           | 0                | 0.0 %         | 29.67      |
>
> Self-review achieves 0-18% recovery, and this relatively low recovery rate comes at high computational costs (26-64 steps). The result shows that enabling the agent to periodically check the home state before the scheduled time and immediately after the scheduled execution does not lead to a significant improvement in task success rates. Hence, the low performance of models on QT4 is likely attributed to the inherently limited capabilities of these models to solve complex tasks based on the home state, rather than to the number of assessment opportunities provided.
>
> **[Experiment 2: Dynamic Re-evaluation with Post-Execution Failure Notice]**
> **Settings:** We implemented a multi-turn feedback loop to test post-execution re-evaluation when a failure is **explicitly reported**. At the scheduled execution time, the SimuHome simulator checks whether the target device state was achieved. If not, a user simulator (GPT-5-mini) provides natural language feedback to the agent (e.g., "The device you scheduled is not in the expected state"). The agent then re-attempts the task with this feedback. It is important to note that this setting is quite unrealistic and highly favorable to the agent, because in real scenarios, it is difficult to expect an oracle to immediately notify the agent of an execution failure at the scheduled time. Instead, the agent is expected to inspect the success or failure of the scheduled task without the involvement of an oracle or the user, and to fix any failures as in Experiment 1.
> **Results:**
> | Query Type | Failed Cases | Recovery Success | Recovery Rate | Avg. Steps |
> |------------|--------------|------------------|----------------|------------|
> | QT4-1      | 25           | 15               | 60.0          | 6.7        |
> | QT4-2      | 27           | 15               | 55.6%          | 4.7        |
> | QT4-3      | 33           | 22               | 66.7%          | 4.4        |
>
> Compared to the self-review experiment (Experiment 1), post-execution failure notice enables recovery in 55-67% of failed cases with efficient step counts (4.4-6.7 steps). However, the results should be interpreted with caution and viewed as a topline estimate of model performance, because this setting is quite unrealistic, as failure notices require the involvement of an oracle. For users, the initial failure is still perceived as a failure, even if the model can correct it after the user's complaint.
>
> **We have incorporated these two experiments into Section 5.2.4 and Appendix G of the revision.**

---

> > ### Comment · Reviewer_qKgg · 2025-11-27
> >
> > Thank you for your response. Based on my overall assessment of the paper, I will maintain my current positive score.

---

> > > ### Author Response · Authors · 2025-12-03
> > >
> > > Thank you for maintaining your positive assessment and for the constructive feedback that strengthened our work.

---

### Author Response · Authors · 2025-12-03
**Review Summary**

Dear Area Chair,

We deeply appreciate you taking on the additional responsibility of evaluating our submission under these exceptional circumstances.
Our comprehensive rebuttal has successfully addressed **all reviewer concerns** through extensive new experiments, resulting in significant positive outcomes.

&nbsp;

### **[Final Reviewer Assessments Immediately Before the Reviews Were Reverted]**
| Reviewer | Rating | Comments |
| :--- | :--- | :--- |
| **hnkK** | 4 → **8** (Highly raised) | *"... added experiments show some interesting results that are worth further exploration… I've raised my score accordingly"* |
| **wJtQ** | **8** (Maintained) | *"High fidelity... benchmark design is excellent… Analysis on temporal reasoning is brilliant… maintain positive score"* |
| **qKgg** | **6** (Maintained) | *"Smart home control addresses real-world challenges… practically valuable… maintain positive score"* |
| **zEmB** | **6 (No discussion)** | *"A useful artifact for future work… insightful look into model's behavior"* |

&nbsp;

### **[Resolution of Reviewer Concerns]**
**Generalization across Models**
Reviewers hnkK and qKgg questioned generalization across model scales and types. We conducted additional experiments with small models (Llama3.2-1B-it, Llama3.2-3B-it, Gemma3-4B-it, achieving near-zero performance) and advanced reasoning models (Gemini-2.5-Pro, GPT-5.1). Following the same patterns in our original experiments, the small models failed to infer latent user intent, perform temporal reasoning, and detect logical contradictions. Reasoning models showed superior accuracy but prohibitively long inference times (GPT-5.1: 135s vs GPT-4.1: 30s). This makes them impractical for real-world deployment.

**Multi-turn Interaction**
Reviewer qKgg suggested that multi-turn interaction might improve performance. We extended our framework to support users providing clarifications and additional context across turns, observing modest gains (44% → 50-54%). This demonstrates two key insights. First, our benchmark is extensible to multi-turn scenarios and can evaluate interaction-based approaches easily. Second, the modest improvement reveals that models need stronger reasoning capabilities for disambiguation and self-refinement, rather than just more interaction opportunities.

**Runtime Adjustment**
Reviewer qKgg raised concerns that requiring upfront scheduling without runtime adjustment artificially constrains performance. We ran additional experiments where we enabled agents to check and modify schedules every 5 minutes before and after execution. Recovery rates remained low (0-18%) at high computational costs (26-64 steps). This reveals that dynamic re-planning provides limited benefit without strong temporal reasoning.

**Architectural Simplicity**
Reviewers zEmB and wJtQ questioned whether ReAct's simplicity unfairly disadvantaged temporal reasoning. We ran additional experiments using HiAgent, which uses hierarchical memory and advanced planning. Results showed that neither HiAgent nor ReAct consistently succeeded across all temporal reasoning queries. This confirms that the primary challenge lies in the limited temporal reasoning capability of contemporary models, rather than their architectural sophistication.

**Benchmark Difficulty**
Reviewer wJtQ questioned whether our fixed benchmark was easy enough to be solved by memorizing the 12 query types, rather than requiring genuine temporal reasoning. We conducted additional supervised fine-tuning experiments where we trained Gemma3-4B-it on 204 gold trajectories and evaluated it. Despite explicit training on successful trajectories, the performance on temporal reasoning tasks (QT4) remained near-zero, and models failed to generalize to dynamic environmental variations. This demonstrates that our benchmark requires more than simple memorization of static patterns.

&nbsp;

We humbly believe that we have faithfully and extensively run all the additional experiments requested by the reviewers, demonstrating SimuHome as a valuable contribution to agent research that provides both a realistic benchmark and critical insights into current limitations. We respectfully ask that you consider the rigor of our work, our thorough responses to all reviewer concerns, and the supportive assessments reflected in the reviewers' final scores prior to the review reversion.

---

### Meta-Review · Area_Chair_TcxV · 2026-01-09

**Summary:**

This submission introduces SimuHome, a high-fidelity simulator and benchmark designed to evaluate Large Language Model (LLM) agents within smart home environments. Built upon the industry-standard Matter protocol, the simulator supports 17 device types and models environmental variables to create realistic feedback loops. The accompanying benchmark consists of 600 episodes across twelve query types, specifically testing capabilities such as latent intent inference, device control, and complex temporal scheduling. Extensive evaluations of varying agent sizes, including state-of-the-art models like GPT-4.1 and GPT-5.1, reveal that current LLMs struggle significantly with temporal reasoning and maintaining logical consistency in "infeasible" scenarios, highlighting a critical gap between current capabilities and real-world deployment requirements.

**Reviewer Concerns:**

The reviewers initially raised valid concerns regarding the comprehensiveness of the model evaluation, the realism of the interaction settings, and the limitations of the chosen baseline framework. Specifically, reviewers hnkK and qKgg noted a lack of model diversity, requesting evaluations of both smaller, resource-constrained models (<7B parameters) suitable for on-device use and newer reasoning models like GPT-5. Additionally, reviewers qKgg and zEmB questioned the single-turn nature of the benchmark and the reliance on the ReAct framework, suggesting that multi-turn interactions or memory-augmented architectures might mitigate the observed failures in temporal scheduling. Reviewer wJtQ also raised questions about the complexity of environmental interactions and the potential for models to memorize the fixed benchmark episodes.


The authors addressed these concerns comprehensively in the rebuttal. They expanded the evaluation to include small models (Llama3.2, Gemma3) and advanced reasoning models (GPT-5.1, Gemini-2.5-Pro), demonstrating that while reasoning models perform better, they incur prohibitive latency costs. To address interaction realism, they implemented multi-turn experiments and runtime plan adjustments, finding that while these methods offered modest improvements, the fundamental capability gap in temporal reasoning persisted. Furthermore, they compared ReAct with HiAgent (a hierarchical memory framework) and conducted Supervised Fine-Tuning (SFT) experiments, proving that the low performance stems from intrinsic model limitations rather than the architectural framework or simple task memorization. I believe the concerns regarding evaluation scope and architectural validity have been fully addressed, confirming the benchmark's difficulty is well-founded.

**Reviewer Scores:**

Based on the post-rebuttal discussions and final ratings provided in the system, the reviewers have converged on a positive consensus. Reviewer hnkK significantly raised their score from 4 to 8, citing the valuable additional experiments and interesting results. Reviewer wJtQ maintained a strong score of 8, praising the benchmark design and the insightful analysis of temporal reasoning failures. Reviewer qKgg maintained a positive score of 6, acknowledging the practical value of the work despite remaining limitations in real-time constraints. Reviewer zEmB did not participate in the final discussion but initially provided a score of 6, viewing the simulator as a useful artifact.

---

### Decision · Program_Chairs · 2026-01-26

Accept (Oral)